# Single-molecule localization microscopy reveals the ultrastructural constitution of distal appendages in expanded mammalian centrioles

Ting-Jui Ben Chang [1,2,3], Jimmy Ching-Cheng Hsu[2] & T. Tony Yang [2,4] ✉

Distal appendages (DAPs) are vital in cilia formation, mediating vesicular and ciliary docking to the plasma membrane during early ciliogenesis. Although numerous DAP proteins arranging a nine-fold symmetry have been studied using superresolution microscopy analyses, the extensive ultrastructural understanding of the DAP structure developing from the centriole wall remains elusive owing to insufficient resolution. Here, we proposed a pragmatic imaging strategy for two-color single-molecule localization microscopy of expanded mammalian DAP. Importantly, our imaging workflow enables us to push the resolution limit of a light microscope well close to a molecular level, thus achieving an unprecedented mapping resolution inside intact cells. Upon this workflow, we unravel the ultra-resolved higher-order protein complexes of the DAP and its associated proteins. Intriguingly, our images show that C2CD3, microtubule triplet, MNR, CEP90, OFD1, and ODF2 jointly constitute a unique molecular configuration at the DAP base. Moreover, our finding suggests that ODF2 plays an auxiliary role in coordinating and maintaining DAP nine-fold symmetry. Together, we develop an organelle-based drift correction protocol and a two-color solution with minimum crosstalk, allowing a robust localization microscopy imaging of expanded DAP structures deep into the gel-specimen composites.

Centriole contributes to multiple critical cellular functions. It plays a vital role in the cell cycle by forming spindle fibers and initiating the cell cycle[1]. The mother centriole is uniquely responsible for ciliogenesis as the basal body of a primary cilium[2-5]. The primary cilium, protruding from the distal end of the basal body, functions as a sensory hub to mediate the transduction of diverse signals[6]. Distal appendages (DAPs) and subdistal appendages (sDAPs) are present at the distal end of the mother centriole. DAPs are shaped as ninefold symmetric, pinwheel-like structures protruding from the mother centriole. DAPs are essential for ciliary vesicle docking during ciliogenesis[7-10] to regulate axoneme growth and represent a part of the ciliary gate[11-13], while sDAPs serve a role in microtubule (MT) anchoring[14]. Numerous DAP proteins have been identified in mammalian cells for their roles in ciliogenesis and ciliary-associated regulations[10,15-19]. To initiate DAP assembly, C2CD3 is recruited to the centriole-distal end[18,20] and MNR, FOPNL, CEP90, and OFD1 are indispensable for further recruitment of CEP83, CEP89, and SCLT1[21,22]. SCLT1 is then required for FBF1 and CEP164 recruitment[10].

[1]Department of Physics, National Taiwan University, Taipei, Taiwan. [2]Department of Electrical Engineering, National Taiwan University, Taipei, Taiwan. [3]Nano Science and Technology Program, Taiwan International Graduate Program, Academia Sinica and National Taiwan University, Taipei, Taiwan. [4]Graduate Institute of Biomedical Electronics and Bioinformatics, National Taiwan University, Taipei, Taiwan. ✉e-mail: tonyyang@ntu.edu.tw

Besides, ODF2, an sDAP-associated protein, has been suggested for its dual function across sDAPs and DAPs[23–25].

Due to the spatial complexity of appendage structures, investigating such molecular arrangements becomes a significantly challenging task. Superresolution (SR) microscopy, especially single-molecule localization microscopy (SMLM), has been employed for uncovering protein localizations of DAPs and sDAPs with a spatial resolution of ~20 nm[25–29]. Our previous work used direct stochastic optical reconstruction microscopy (dSTORM) to map the architecture of mammalian DAPs and revealed the structure of the DAP matrix between adjacent blades[28]. A further study using correlative SMLM and electron microscopy (EM) demonstrated a close relationship between DAP protein localization and its electron-dense micrograph[29]. However, the resolving power of localization microscopy is still more than an order of magnitude worse than that of EM, which thus results in different structural interpretations between these two scopes. Moreover, CEP83 was previously reported to own a minimum measure among the outer DAP structure[28]. Recent studies further indicate that MNR, FOPNL, CEP90, and OFD1 are found near the DAP base between CEP83 and the microtubule triplet[21,22]. Nevertheless, their ultrastructural arrangement at a molecular level remains to be investigated. While the superresolution imaging suggested that C2CD3 is concentrated in the centriole lumen[28,30,31], an apparent spatial relationship between C2CD3 and other core DAP proteins is still unclear due to insufficient technical evidence based on the given mapping resolution. These imply that we urgently need an advanced imaging modality with an adequate spatial resolution for investigating the molecular configuration of DAPs.

By embedding cellular structures into the network of swellable polyelectrolyte hydrogel, one can physically expand a biological specimen to enable SR imaging under conventional fluorescence microscopy—referred to as expansion microscopy (ExM)[32–34]. Numerous expansion strategies have been proposed for different purposes, for example, magnified visualization of protein, RNA, and transcriptomics in various specimens: cells, isolated centrioles, tissues, and brains[35–39]. Further resolution enhancement can be achieved by integrating ExM with structured illumination microscopy (SIM)[40], stimulated emission depletion microscopy (STED)[41], or SMLM[42,43]. Notably, a combination of ExM and localization microscopy enables us to push the resolution limit of a light microscope to a molecular level. Nonetheless, the increased distance between organelles and coverslips makes drift correction an arduous task, limiting us to finding proteins of interest only near the coverslip. The deficiency of a pragmatic imaging solution and drift correction protocol also hinders the benefits of multi-color imaging in practice. Although single-molecule localization imaging of expanded samples ideally demonstrates its ability to resolve the ultrastructural features, general applications to a broad spectrum of biological questions remain challenging.

In this study, we strategically integrate ExM with dSTORM (Ex-dSTORM) to elucidate ultrastructural details of mammalian DAP structure in intact cells. First, we proposed a practical workflow incorporating in situ drift correction to enable Ex-dSTORM imaging throughout the entire cell, not limited to coverslips. Second, we optimized the combination of red and far-red dyes for yielding minimum spectral crosstalk in two-color Ex-dSTORM. Therefore, using Ex-dSTORM, we reached unprecedented details in understanding DAPs and gained extensive insight into their three-dimensional (3D) ultra-construction of the higher-order protein complexes. We uncovered the protein organization at the DAP base and bridged the spatial relationship between DAPs and centriole to interpret the 3D computational model precisely. Finally, our Ex-dSTORM study elucidated how ODF2 affected the distal appendage at the structural level and suggested its role in DAPs.

## Results

### In situ drift correction enables systematical molecular-resolution protein mapping with optimized Ex-dSTORM

To systematically investigate the ultrastructural details and protein-protein spatial relationship of DAPs in mammalian cells, we optimized the Ex-dSTORM from several different perspectives (Fig. 1). In our experiment, we prepared samples with a post-labeling expansion procedure. It is evident that this sample preparation offers many advantages: high labeling density, reduced linkage error, and reduced antibody competition. Moreover, the monomer concentration used in the post-labeling ExM could affect the ultrastructural context of multiprotein complexes[39]. The workflow of post-labeling ExM is shown in Fig. 1a. The optimized perfusion concentration in our experiment (1.4% FA and 2% AA in 1× PBS) can achieve a greater expansion factor and, thus, a higher resolution for DAP structure (Supplementary Fig. 1). Special note is given to the re-embedding process. Compared with a previous study, where a 20-30% shrinkage after re-embedding was found[36,43], we optimized the protocol to retain the re-embedded hydrogel at the approximately original size. Our process using the re-embedding reagents (10% AA, 0.15% BIS, 0.05% APS and 0.05% TEMED) in only ddH$_2$O−rather than in Tris buffer−with no bind-silane coating could hold ~94.5% of the original size of the expanded hydrogel after re-embedding (Supplementary Note 1 and Supplementary Fig. 2). Thus, we could obtain nearly 4-fold linear expansion after re-embedding.

The lateral drift is an inevitable issue in the SMLM because of the long image acquisition, and consequently, a post-processing correction is regularly needed. A specimen expansion procedure usually brings cellular organelles tens of micrometers away from the coverslip. More specifically, fiducial beads (for example, gold beads and microspheres) frequently used in the SMLM are often deposited on the coverslip surface. They are presumably outside the detection focus at the cellular structure of interest, thus causing difficulties in the operation of drift correction (Fig. 1b).

Here, we conceived an image-based method for in situ lateral drift correction by monitoring cellular structures (marker protein) adjacent to the proteins of interest (target protein) (Supplementary Note 2 and Supplementary Fig 3a). In order to perform the dSTORM imaging across the expanded cellular structures, we adopt the epi-illumination in our optical system (Fig. 1b). During image acquisition of the target protein, we also allow the images of marker proteins to be intermittently recorded for following pattern correlation analysis (Fig. 1c and Supplementary Fig. 3b). We utilized ATP synthase, for example, as the marker protein, which is enriched and widely distributed over a cell (Supplementary Fig. 3a). With the marker's feature occupying most of the cell, one can readily locate cells within the expanded gel and then look for the target protein nearby. To our satisfaction, our method no longer constrains the study of cellular structures to merely the coverslip surface. Besides, the intense fluorescence signals from ATP synthase allow samples to endure frequent exposure to the laser light, improving the accuracy of drift correction and contributing to high localization precision. Therefore, the in situ drift correction allows us to obtain dSTORM images of expanded cells with high fidelity even the targets are far from the coverslip (Supplementary Fig. 3b). Since this approach employs a common detection channel (green) to record marker images, the additional chromatic aberration usually arising from different colors of markers can be avoided. Given that the marker protein is within the hydrogel, any relative movement between markers and targets could be minimized.

Furthermore, we customized a mounting chamber for Ex-dSTORM (Supplementary Fig. 4). This design provides excellent mechanical stability and ensures that samples are immobilized in the chamber to minimize lateral drift between the cover glass and the sample. Therefore, we only need to correct the drift between the objective and the samples. Lastly, we estimated the system resolution

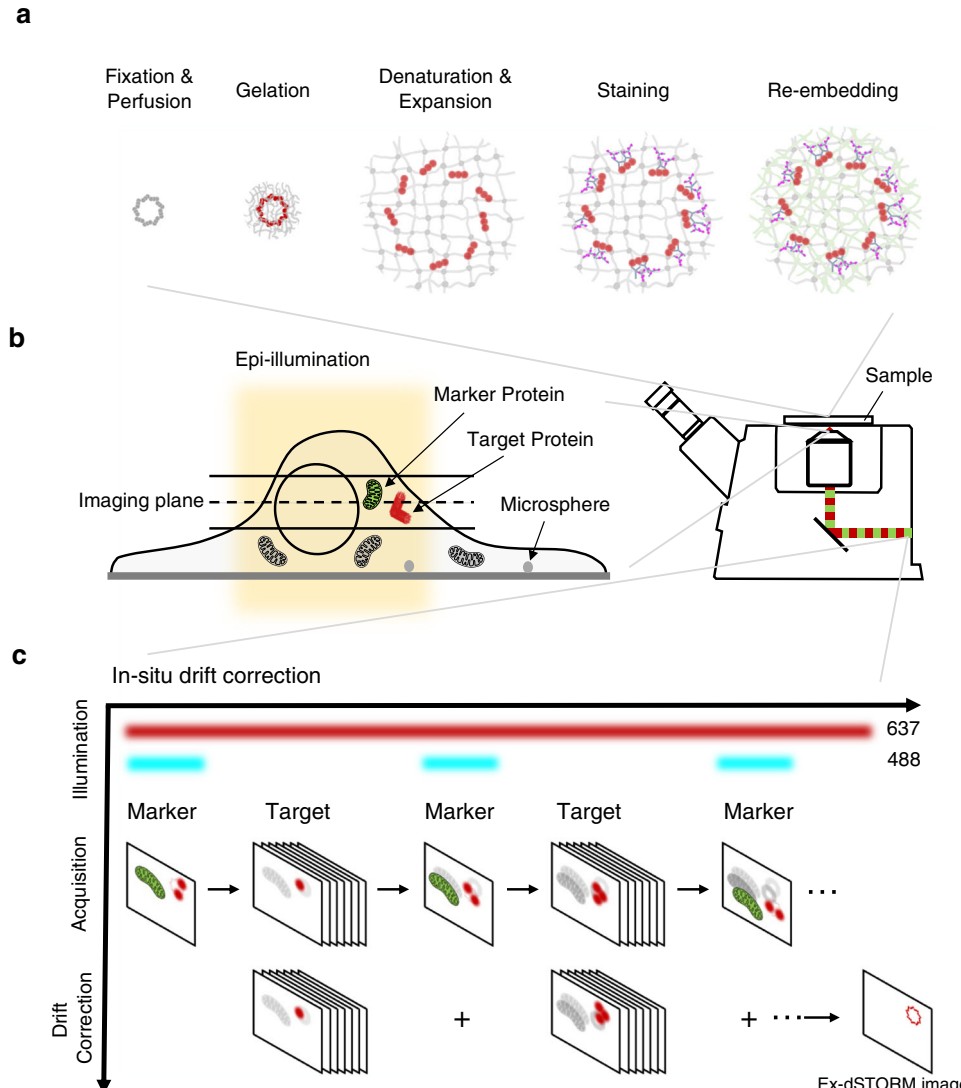

**Fig. 1 | Schematics of optimized Ex-dSTORM with in situ drift correction.**
**a** Sample preparation utilizing an optimized post-labeling ExM, including fixation, perfusion, gelation, denaturation, expansion, staining, and re-embedding. Both optimized perfusion concentration and re-embedding solution without Tris were used for a higher expansion factor (details in Methods and Supplementary Note 1). **b** In situ marker labeling enabling in situ drift correction with Ex-dSTORM imaging. The microsphere beads deposited on the coverslip fail to be the fiducial markers in the expanded sample due to a long distance from the imaging plane at the target protein. Epi-illumination is adopted to excite the expanded cells. **c** Schematic of the

in situ drift correction. Through immunolabeling marker proteins across cells in (**b**) (green channel), Ex-dSTORM imaging of target proteins can be performed anywhere inside the expanded gel. Besides the 637-nm excitation light for single-molecule imaging, we intermittently light a 488-nm laser to acquire the lateral drift of the sample for post-processing position compensation. A smooth interpolation between two marker image series is performed owing to slow drifting. Then, the resulting Ex-dSTORM image with no markers is obtained by superimposing the localizations of single molecules.

of our dSTORM by first performing the Fourier ring correlation (FRC) analysis[44,45] (Supplementary Fig. 5). Then we further divided the system resolution by the average expansion factor to yield the shortest distance between two labeled molecules that can be distinguished with Ex-dSTORM, i.e., ~5.79-nm (Supplementary Fig. 5). The resolving power is ideal for investigating the ultrastructure of protein complexes with quantification analysis at a molecular scale.

### Revealing ninefold symmetry of C2CD3 and ultrastructural contexts of DAP proteins with Ex-dSTORM

Superresolution microscopy has been used to image many centriolar proteins, but studies on ultrastructural contexts of DAPs have not yet been reported due to insufficient resolution. Hence, we systematically imaged the core DAP proteins (CEP83, CEP89, SCLT1, FBF1, and CEP164), a centriole-distal-end protein (C2CD3), and a centriolar

structure protein (Ac-tub) in human retinal pigment epithelial cells (RPE-1). First, we synchronized RPE-1 cells in the $G_1/G_0$ phase to arrest the spatial arrangement of these proteins in the fully-grown state. Then, we conducted Ex-dSTORM imaging of them from the axially oriented mature centrioles by searching for circular ring patterns of protein images. A remarkable improvement in resolution is found in the Ex-dSTORM results compared to other imaging methods (Fig. 2a). Notably, the ultrastructural arrangement of DAP proteins at individual blades can only be revealed by Ex-dSTORM. To further determine the organelle-defined expansion factor for a more accurate quantitative analysis, we divided the mean diameter of SCLT1 analyzed from Ex-dSTORM images by that analyzed from dSTORM images (Supplementary Table 1). This analysis yields an expansion factor of 3.92, slightly smaller than the value obtained from a direct measurement of gel size. Surprisingly, C2CD3, a protein essential for DAP assembly,

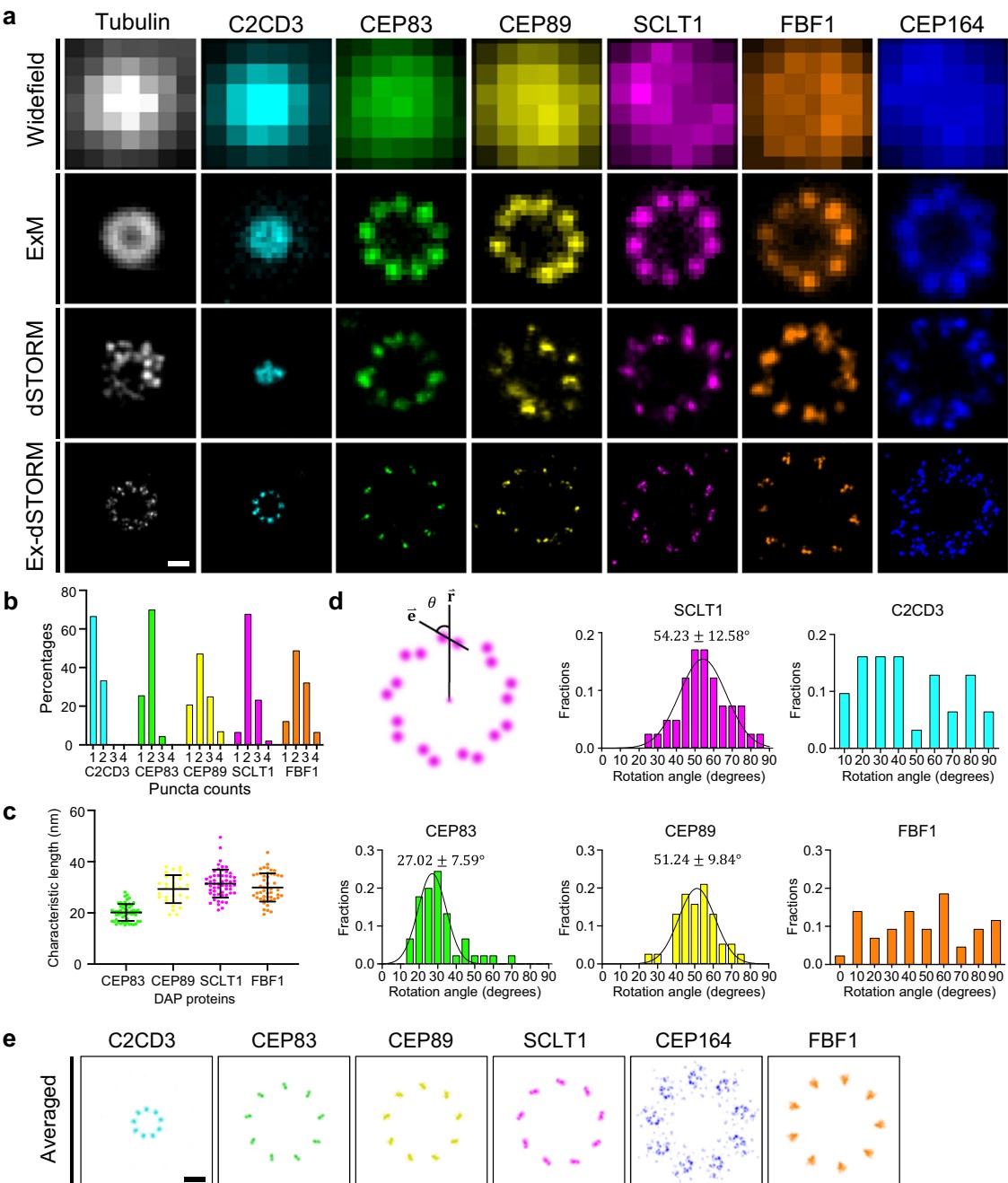

**Fig. 2 | Ultrastructural protein mapping of distal appendage (DAP) and C2CD3 achieved by Ex-dSTORM. a** Representative images of various DAP and its associated proteins under different microscopy techniques. With the ~40× resolution enhancement, Ex-dSTORM exhibits ultrastructural details of the DAP proteins. **b** Histogram analysis of the number of puncta discovered at each blade of DAP proteins. Most DAP proteins show more than one punctum in each blade, forming rod-like patterns ($n = 72$ blades for CEP89; $n = 90$ blades for other proteins). **c** Characteristic length analysis of DAP proteins with rod-like patterns in (**b**) except C2CD3, which mostly appears as one punctum in a single blade (mean ± SD, $n = 58$, 32, 55 and 49 blades CEP83, CEP89, SCLT1, and FBF1, respectively). This length indicates the 2D axial-view projection of the 3D arrangement of these proteins. **d** Statistical analysis of the rotation angle ($\theta$) of DAP proteins with rod-like patterns observed from the distal end. SCLT1, CEP83, and CEP89 exhibit the specific anticlockwise rotation angles ($\theta$) between the radial direction ($\vec{r}$) and the characteristic orientation ($\vec{e}$), while FBF1 and C2CD3 show a dispersed distribution in angle ($n = 31$, 45, 38, 41 and 43 blades for C2CD3, CEP83, CEP89, SCLT1 and FBF1, respectively). **e** Averaged images of DAP proteins enhancing corresponding localization features. One representative Ex-dSTORM image acquired from nearly top view for each protein is processed as described in Supplementary Fig. 7 to obtain the averaged image. In particular, the averaged image of FBF1 demonstrates its possible localizations rather than a specific pattern. Scale bars, 100 nm (**a**, **e**). Source data are provided as a Source Data file.

presents as an apparent ninefold radial symmetry under Ex-dSTORM (Fig. 2a), which is different from the previous findings that suggested either a concentrated distribution[28,30,31] or the asymmetric pattern[46] in the centriole lumen. Instead, C2CD3 may show a close spatial correlation with the centriole wall (Ac-tub). The discrepancy between our images and the previous finding showing the asymmetric distribution

may result from the localization variation within nine C2CD3 puncta (Supplementary Note 3 and Supplementary Fig. 6).

For some DAP proteins, our Ex-dSTORM images revealed multiple localization puncta at each blade and organized patterns in these puncta (Fig. 2a–d). Two isolated puncta at individual blades are mainly found for SCLT1, CEP83, and CEP89 (Fig. 2b). Their different

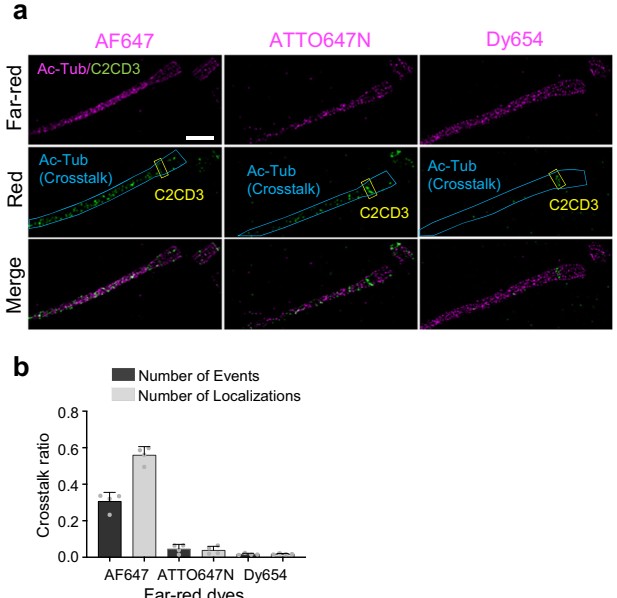

**Fig. 3 | Crosstalk analysis among far-red dyes. a** Two-color Ex-dSTORM images of C2CD3 labeled with CF568 and Ac-Tub stained with different far-red dyes, displaying a crosstalk effect in the red detection channel. Yellow boxes mark the signals of C2CD3, while blue lines enclose the crosstalk signals from Ac-Tub staining with AF647, ATTO647N, or Dy654. **b** Quantitative analysis of crosstalk ratio among these far-red dyes with two analytical methods based on single-molecule events or localizations. Both results indicate the minimum crosstalk using Dy654 (mean ± SD, $n = 4$ cells for each far-red dye). Scale bar, 500 nm (**a**). Source data are provided as a Source Data file.

characteristic lengths (XY-plane projection) may indicate distinct 3D orientations of the DAP proteins (Fig. 2c). To determine whether the orientation of puncta is conserved, we also performed statistical analysis by characterizing the angle from the radial direction (Fig. 2d). Our results suggest the specific rotation angles of -27°, -51°, and -54° (anticlockwise) for CEP83, CEP89, and SCLT1, respectively (Fig. 2d). Interestingly, these three proteins displayed the same anticlockwise chirality different from MT triplets when viewed from the distal end. In contrast, FBF1 and C2CD3 puncta presented a broad angular distribution. As for CEP164, the proteins showed a clear clockwise chirality and formed a curved feather-like shape (Fig. 2a).

Lastly, we performed a rotational average of the localization signals based on a ninefold symmetric configuration to feature the ultrastructural orientation among the DAP components (Supplementary Fig. 7). The DAP proteins were organized in certain chirality except for FBF1 (Supplementary Table 2), implying that FBF1 may involve different ciliary mechanisms other than being a structural component of DAPs (Fig. 2e).

### Optimized dye combination for minimum crosstalk in two-color Ex-dSTORM

Spatial relationships among distinct proteins could not be revealed by single-color imaging. Nevertheless, due to several practical challenges, two-color Ex-dSTORM has not been routinely applied to study biological questions. To this end, we optimized two-color Ex-dSTORM with red and far-red channels, which generally demonstrate excellent optical properties for single-molecule signals.

First, Alexa Fluor 647 (AF647), a frequently used dye in two-color SMLM, was used for the far-red channel, and CF568 was used for the red channel. In our experiment, we observed an abundance of additional signals—other than the signals from CF568—in the red channel detection. To this, the observed event is essentially a crosstalk effect.

Furthermore, to quantify the crosstalk phenomenon and search for the slightest crosstalk among far-red dyes, we immunolabeled two proteins, C2CD3 with CF568 (which only appeared on the centriole-distal-end, yellow box, Fig. 3a) and Ac-Tub with three frequently used far-red dyes, AF647, ATTO647N, and Dyomics 654 (Dy654) (Fig. 3a). This protein combination allowed us to recognize crosstalk at the cilium compartment because of their different spatial distribution and protein numbers.

In the far-red channel, we only observed the pattern of Ac-Tub under our expectations. Nonetheless, in the red channel, besides the designated pattern of C2CD3 (yellow box, Fig. 3a) we also detected the pattern of Ac-Tub (regions marked with blue lines, Fig. 3a). We found Dy654 demonstrated the most negligible crosstalk effect in the red channel (below 2% in both analyses) (Fig. 3b). On the contrary, given the observation of Ac-tub, AF647 reached over 30% of the crosstalk effect, which may cause significant image artifacts to the red channel. As a result, we chose Dy654 as our far-red dye in the following experiment to provide more reliable information, especially in two-color Ex-SMLM. Hence, this dye combination allows us to accomplish two-color molecular-resolution protein mapping with high fidelity.

### Ex-dSTORM axial-view imaging reveals ultra-resolved angular and radial relationships among DAP proteins

We next performed minimized-crosstalk two-color Ex-dSTORM imaging from axial orientation to redefine the configuration of DAPs at the molecular level. We have made several discoveries. Firstly, relative allocations of proteins in the radial direction revealed the protein-protein spatial relationships. The proteins C2CD3, SCLT1, CEP83, CEP89, and CEP164 were imaged and paired. From our result, the two-color image pairs of C2CD3-SCLT1, CEP83-SCLT1, CEP83-CEP89, and CEP164-SCLT1 demonstrated that they were all arranged in the same radial direction (Fig. 4a–c, e). By contrast, the image pair of FBF1-SCLT1 showed that FBF1 was in a different radial direction among DAPs (Fig. 4d). Moreover, we found that CEP83 partially overlapped with CEP89 and that, statistically, CEP83 displayed a slightly smaller radius (Fig. 4c). We also found that SCLT1 was nearly at the centroid of CEP164 patterns and enclosed by the clockwise feather-like signals of CEP164 (Fig. 4e).

Secondly, we performed quantitative analyses from the axial orientation to unveil the protein-protein angular relationships—angular spacing and relative angle. For starters, the angular spacing of all protein pairs was measured and found to be precisely repeated every 40° interval (Supplementary Figs. 8, 9). Notably, histograms of angular spacing from our Ex-dSTORM images demonstrated a much more satisfactory angular resolution of about 0.5°, crucial for differentiating spatial arrangement among DAP proteins at a few nanometers. Besides angular spacing, we explored relative angles from the histogram analyses using SCLT1 as the reference. The centers of C2CD3, CEP83, CEP89, and CEP164 were arranged at a concentrated angle of around 0° but with distinct distributions (Fig. 4f; Supplementary Fig. 10). As for the relative angle of FBF1, on the contrary, we found that it localized at a distinctive angle of ~12.9° with respect to SCLT1, suggesting that it was not aligned with other DAP proteins. Nevertheless, regular SMLM could have difficulty differentiating between FBF1 and SCLT1 as it does not have enough resolving power[29]. More importantly, this result supports our previous finding that FBF1 may serve a unique role in ciliary gating in addition to DAP formation[28].

Thirdly, we further conducted the radial analysis to investigate the configuration of DAPs. In our result, the radial analysis of C2CD3 revealed neither the disc-like pattern nor concentration in the lumen[28,30,31] (Fig. 4f); this suggests that our Ex-dSTORM can outperform typical dSTORM to resolve the ninefold radial pattern. Besides, it is worth mentioning that we eliminate the cell-to-cell difference and blade-to-blade variation for the radial analysis by measuring the individual radius ratio blade by blade (Supplementary Fig. 11).

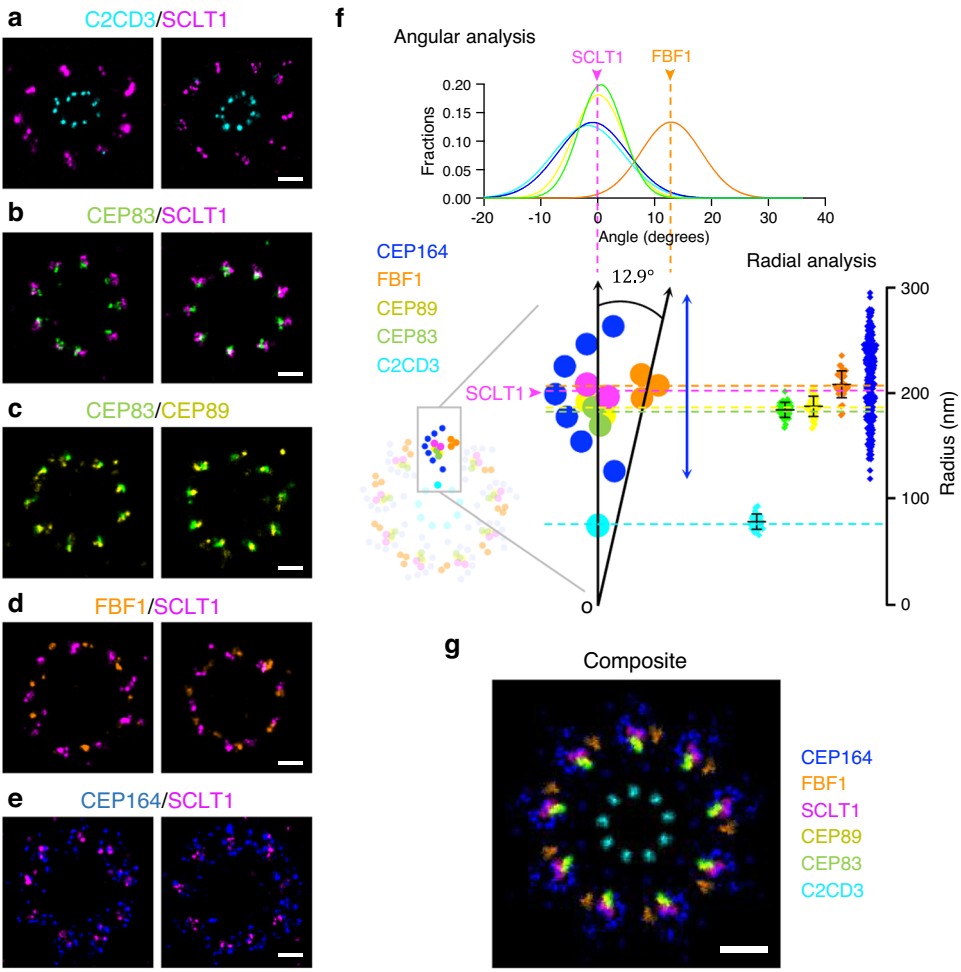

**Fig. 4 | Revealing ultra-resolved angular and radial relationship among DAP proteins and C2CD3 with two-color Ex-dSTORM imaging. a–e** Two representative two-color axial-view Ex-dSTORM images. C2CD3-SCLT1 (**a**), CEP83-SCLT1 (**b**), CEP83-CEP89 (**c**), and CEP164-SCLT1 (**e**) pairs align in the same radial direction, whereas FBF1 exhibits a distinct angular position with respect to SCLT1 (**d**). SCLT1 is enclosed by the "feather-like" CEP164 distribution (**e**). **f** Angular and radial analyses of DAP proteins. The angular analysis demonstrates that C2CD3, CEP83, CEP89, and SCLT1 allocate at a concentrated angular distribution around 0° (*n* = 243 points for each pair), whereas the mean position of FBF1 deviates from SCLT1 by 12.9°. Radial

analysis reveals the mean radius of C2CD3, CEP83, CEP89, SCLT1, and FBF1 (mean ± SD, *n* = 34 blades for FBF1/SCLT1; *n* = 36 blades for other pairs), while CEP164 proteins are represented with their individual localizations (*n* = 261 points from 4 centrioles). **g** Composite of averaged images of DAP proteins and C2CD3 (Fig. 2e) scaled with their mean angular and radial positions. FBF1 is exceptionally located in a different radial direction in contrast to CEP83, CEP89, and SCLT1 enclosed by CEP164. Scale bars, 100 nm (**a–e**, **g**). Source data are provided as a Source Data file.

---

Taken together, we reconstructed the ultrastructural composite image by assembling the information of radius ratio, relative angle, and averaged images (Fig. 4g). Surprisingly, although most DAP proteins appeared in the anticlockwise direction, the resulting profile in the composite represented the clockwise direction.

### Ultra-detailed analyses of the relative longitudinal relationship among DAP proteins

Using SCLT1 as the reference, we obtained a relative longitudinal position among DAP proteins. Our result denoted a few intriguing features of these DAP proteins from the lateral view with a series of two-color Ex-dSTORM imaging. First, CEP83 possessed two-layered populations separating further in the longitudinal direction; moreover, CEP83 constructed a colinear correlation with SCLT1 (gray lines, Fig. 5a). CEP89, SCLT1, and FBF1, in contrast to CEP83, mainly exhibited nearly horizontal distributions and formed a certain inclination angle (dashed lines, insets, Fig. 5a). Second, to underline the stereoscopic sense of DAPs structure, we further captured two-color Ex-dSTORM images from a tilted view. A combination of CEP83-SCLT1 revealed the crown-like configuration (tilted view, Fig. 5a). While CEP83-CEP89

showed similar values in radius, they manifested an explicit offset in the longitudinal orientation compared to the axial observation (tilted view, Figs. 5a, 4c). Third, we found that most inner molecular localizations of CEP164 presented at lower longitudinal positions (arrowheads, Fig. 5a). In addition, we confirmed that SCLT1 was located within the geometric center of CEP164 as if a curve feather of CEP164 was embracing SCLT1 (tilted view, Fig. 5a).

To quantify the longitudinal position of each DAP protein, we again used SCLT1 as the reference to determine each localization of molecular puncta. Ex-dSTORM imaging enabled us to perform the paired measurement in both the axial and lateral view (Supplementary Fig. 11). Therefore, we could rule out the longitudinal position variation from different blades, obtaining an accurate molecular position of each protein, not merely reporting an average position (Fig. 5b). Histogram analysis disclosed the two-layered CEP83 distribution, which was separated by ~26.0 nm (green arrowheads, Fig. 5b). The histogram analysis also revealed a broad distribution of CEP164 along the longitudinal direction (blue arrows, Fig. 5b). For molecular orientation, the inclination angle of CEP83, CEP89, SCLT1, and FBF1 with ~52.1°, ~21.0°, ~16.4°, and ~19.5° can be determined from axial (*XY*) and lateral (*Z*)

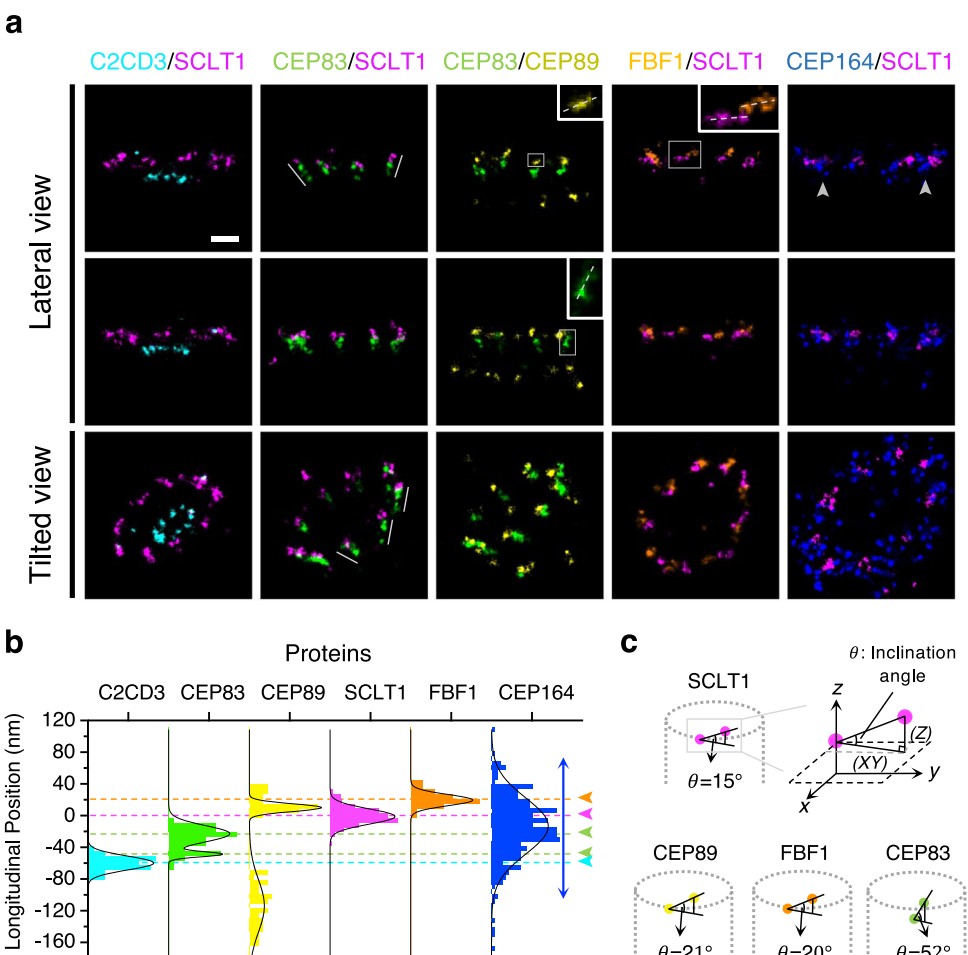

**Fig. 5 | Lateral-view Ex-dSTORM images enabling the ultrastructural analysis of the longitudinal position of DAPs and C2CD3. a** Representative two-color Ex-dSTORM images from a lateral view and tilted view. Each pair of DAP proteins from lateral-view images illustrates their relative longitudinal positions. The collinear relationship is particularly found in the CEP83-SCLT1 images (gray lines). The angle of inclination of each DAP protein is identified for CEP83, CEP89, FBF1, and SCLT1 (dashed lines, insets). Gray arrowheads mark the narrower radial distributions at the lower bounds of CEP164 longitudinal localizations. The tilted-view images highlight the stereoscopic interpretation of the spatial arrangement of DAP proteins. **b** Histogram analysis of the longitudinal positions of DAP proteins and C2CD3 relative to SCLT1 ($n = 7, 9, 14, 24, 13$ and centrioles for C2CD3, CEP83, CEP89, SCLT1, FBF1, and CEP164, respectively). Arrowheads indicate the mean longitudinal position of each protein except for CEP164, which distributes throughout the other proteins (blue double-headed arrow). **c** Computation of the inclination angle of DAP proteins at each blade for SCLT1, CEP89, FBF1, and CEP83 by the ratio of the longitudinal projection ($Z$) to axial projection ($XY$). Scale bar, 100 nm (**a**). Source data are provided as a Source Data file.

projection (Fig. 5c; Supplementary Table 2). In particular, we found that the inclination angle of CEP83 is in good agreement with that of the DAP blades from the EM micrographs[26,29]–this indicated that CEP83 represented a direct structural involvement in the DAP profile.

## Ex-dSTORM reveals the ultrastructural constitution of the DAP base

Previous studies frequently overlapped SMLM with EM images to bridge the spatial relation between DAP and centriole[26,28,29]. Nonetheless, resolutions between the two techniques diverge more than an order of magnitude apart. Hence, it is prone to cause different data interpretations without comparable image quality between different methods[28,29]. Moreover, the root DAP structure near the centriole was unclear due to an insufficient resolution. In order to directly and precisely correlate DAP structure with a centriole, we first imaged Ac-Tub to examine centriolar triplets, which displayed more clearly in the averaged image (Supplementary Fig. 12). Next, we performed two-color Ex-dSTORM imaging of Ac-Tub and C2CD3. Interestingly, our result shows that C2CD3 localized between adjacent triplets, not concentrating in the centriole lumen and that C2CD3 was arranged in

an extension line of triplets (gray lines, Fig. 6a). To determine whether C2CD3 is close to A-tubule or C-tubule end, we quantified the correlation of C2CD3 and Ac-Tub with radial and angular information. We first evaluated the radius ratio of C2CD3 over MT triplets, and the value of 0.733 was obtained (Fig. 6b). To precisely determine the angular relationship between C2CD3 and Ac-Tub, we measured the angle between the center of the triplets and their adjacent C2CD3. Our finding showed that C2CD3 proteins were positioned between triplets near A-tubule ($\theta$ -20.5°, Fig. 6b). Incorporating the information of DAP proteins/C2CD3/Ac-Tub, we illustrated the coverage of these proteins in their radial distributions (Fig. 6c). Notably, because of significant improvement in resolution and accurate labeling sites with Ex-dSTORM, we could better recognize this specific gap between CEP83 and Ac-tub. Although two recent studies have shown that a complex containing MNR, CEP90, OFD1, and FOPNL is found in this gap[21,22], their molecular arrangement remains to be explored. Here, we conducted the Ex-dSTORM imaging of MNR, CEP90, and OFD1 to observe their molecular localization. Besides the clear ninefold symmetric pattern observed from these proteins, their possible chirality has further been revealed (Fig. 6d and Supplementary Table 3). The relationship of

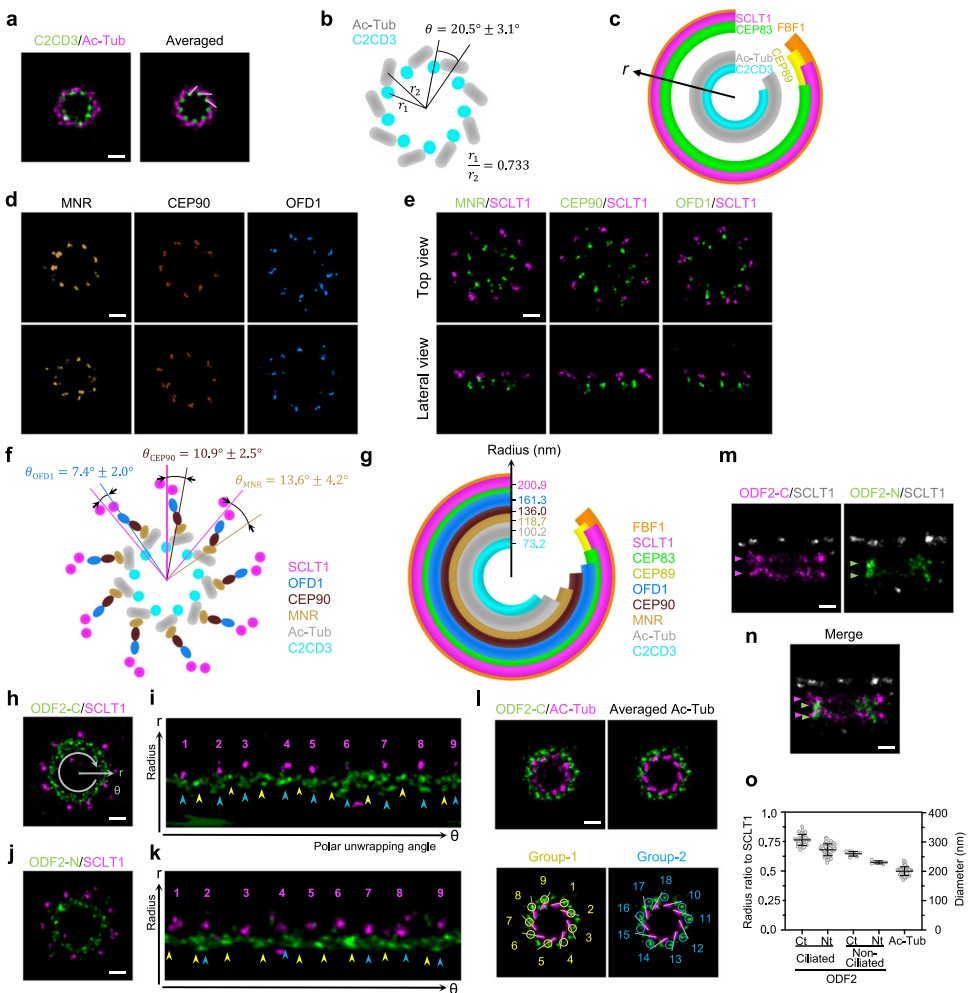

**Fig. 6 | Ex-dSTORM revealing the distinct coverages of DAP proteins at the base. a** A representative two-color Ex-dSTORM image of C2CD3/Ac-Tub disclosing the direct spatial relationship between MT triplets and DAP. The image is further enhanced by rotational averaging every 40° (right). C2CD3 localizes inwards at the extension line of the Ac-Tub (gray lines). **b** Spatial relationship between C2CD3 and Ac-Tub denoting their angular difference and radius ratio ($n = 42$ blades for radial analysis, $n = 4$ centrioles for angular analysis). **c** Radial coverage of the DAP proteins and MT triplets (mean ± 3 SD). **d** Two representative Ex-dSTORM images of DAP-associated proteins MNR, CEP90, and OFD1. **e** Two-color Ex-dSTORM images of MNR, CEP90, and OFD1 with SCLT1 from the top and lateral views. **f** Schematic arrangements of MNR, CEP90, and OFD1 indicating the corresponding quantitative angular displacements against SCLT1 (mean ± SD). **g** Radial coverage marked with mean radii of MNR, CEP90, and OFD1 localizing to the area between CEP83 and MT triplets (mean ± 3 SD). **h, j** Representative two-color Ex-dSTORM images in the axial view of the ODF2-C (**h**) and the ODF2-N (**j**) with SCLT1. **i, k** Polar unwrapping analysis revealing that both ODF2-C and ODF2-N can be characterized into two groups- one localizing at the same angular position ($\theta$) with SCLT1 (blue arrowheads) and the other localizing between SCLT1 (yellow arrowheads). **l** Two-color Ex-dSTORM image of ODF2-C/Ac-Tub manifesting two-grouped populations of ODF2. Signals are enhanced by rotational averaging. Group-1 locates at the extension line of Ac-Tub signals (yellow circles). Group-2 locates between two adjacent MT triplets (blue circles). **m** Representative two-color Ex-dSTORM images in lateral view of ODF2-C and ODF2-N with SCLT1. Two-layered distribution can be clearly distinguished. **n** Merged image from (**m**) for comparing longitudinal position. **o** Quantitative analysis of ODF2-C/N radii for ciliated and non-ciliated cells (mean ± SD, $n = 4, 5, 9, 7$ and 5 centrioles for ciliated ODF2-C and ODF2-N, non-ciliated ODF2-C and ODF2-N, and Ac-Tub, respectively). Scale bars, 100 nm (**a, d, e, h, j, l, m, n**). Source data are provided as a Source Data file.

these proteins in the DAP was also investigated using two-color Ex-dSTORM imaging with SCLT1 as the reference. Intriguingly, MNR shows the opposite chirality to SCLT1, while CEP90, MNR, and SCLT1 have the same chirality (Top-view, Fig. 6e). Based on the quantitative analyses for the radial and angular measure, we could depict the molecular arrangements of these proteins (Fig. 6f). The longitudinal positions of these proteins were also characterized from the lateral view using two-color Ex-dSTORM imaging (Supplementary Fig. 13 and Supplementary Table 4). Together, our results disclose the elaborate molecular organization of MNR, CEP90, and OFD1 in the area between CEP83 and microtubules (Fig. 6g and Supplementary Tabel 3).

On the other hand, we further characterized ODF2, an sDAP component that relates intimately to DAP in terms of its spatial position or biological function[23-25]. To elucidate the configuration and

spatial relationship between ODF2 and DAP, we co-immunostained SCLT1 with ODF2 at its C- or N-terminus. Although two-color Ex-dSTORM images of ODF2-C/SCLT1 and ODF2-N/SCLT1 demonstrated a continuous distribution rather than a ninefold radial pattern as other DAP proteins (Fig. 6h, j) we could still classify ODF2-C and ODF2-N into two different groups of distributions by polar unwrapping analysis. Our result revealed that one group of puncta patterns aligned with SCLT1 with the same angular direction (blue arrows, Fig. 6i, k); another group was positioned between two adjacent SCLT1 (yellow arrows, Fig. 6i, k). To observe the pattern of ODF2 clearly, we also imaged ODF2-C with Ac-Tub from the axial orientation to scrutinize its spatial arrangement with a different reference (Fig. 6l).

Interestingly, two groups of arrangement could be recognized clearly in the ODF2-C with averaged Ac-Tub: one is distributed at the

extension lines of MT triplets (yellow circles, Group-1, Fig. 6l), while the other is localized between two extension lines of MT triplets (blue circles, Group-2, Fig. 6l). We then move forward to image ODF2-C/N from lateral orientation to elucidate the difference between these two. The two-color Ex-dSTORM images from the lateral view revealed that both ODF2-C/N exhibited a two-layered distribution (arrows, Fig. 6m). Moreover, aligning with SCLT1, we found that ODF2-C/N form alternating two-layered distribution beginning with distal ODF2-C layer followed by distal ODF2-N layer, proximal ODF2-C layer, and proximal ODF2-N layer (Fig. 6n). Assembling the information of two groups of ODF2-C/N from axial and lateral views, we inferred that one group of ODF2-C/N from the axial view belongs to the upper layer and the other belongs to the lower layer.

Intriguingly, the quantitative analysis of the ODF2-C/N radii showed that the ODF2 radius of the non-ciliated centrioles was systematically smaller than that of the ciliated cells (Fig. 6o), indicating the ciliation-dependent distribution for ODF2. Given the results from the radial and longitudinal analyses, we found that ODF2 also localizes to the specific coverage near the base of the DAP structure, possibly involved in the DAP constitution (Supplementary Figs. 13, 14; Supplementary Tables 3, 4).

### Ex-dSTORM unravels the specific spatial correlation between the distal-layered ODF2 and DAPs

Longitudinal position analysis demonstrated that the distal-layered ODF2-C/N partially overlapped with DAPs (Supplementary Fig. 13), while the proximal-layered ODF2-C/N was in the region of sDAPs. Moreover, we found a ~18° offset between the aforementioned two groups of ODF2 (Supplementary Fig. 15). Hence, we assumed that the distal-layered ODF2-C/N might be involved in the constitution of DAPs and the proximal-layered ODF2-C/N to be a part of sDAPs. DAPs and sDAP may form in a staggered arrangement stemming from the angle difference between the two groups. To test this hypothesis, we imaged the sDAP-depleted centrioles by studying *CEP128* CRISPR/Cas9 knockout cells[25] to investigate the ODF2 localizations under Ex-dSTORM. Astonishingly, unlike the two-group distributions in WT cells, a canonically ninefold symmetric pattern re-emerged in both ODF2-C and ODF2-N images in the sDAP-deficient cells (arrows, Fig. 7a, b).

The relationship between the remaining layer of ODF2 and DAP was then characterized by two-color Ex-dSTORM (Fig. 7c, d). Strikingly, ODF2-C was allocated in line with SCLT1 (Fig. 7c). Contrarily, ODF2-N was located between the two adjacent SCLT1 (Fig. 7d). In the composite image (Fig. 7e), the radius of ODF2-C is shown to be larger than that of ODF2-N. Moreover, ODF2-C/N interlaced with each other, while ODF2-N manifested a broader distribution in the circumferential direction. Later, with quantitative analysis of the radius ratio in WT and *CEP128*[−/−] cells, we found a systematically smaller radius ratio of ODF2-C/N in *CEP128*[−/−] cells (Fig. 7f). A schematic model for positioning ODF2-C/N and SCLT1 viewed from the distal end is illustrated in Fig. 7g with quantitative data.

To determine how the remaining layer of ODF2-C/N in sDAP-depleted cells is associated with that in the WT cells, we utilize SCLT1 as the reference to analyze the longitudinal position of ODF2-C/N with two-color Ex-dSTORM. Our images showed that the remaining layers of ODF2-C/N in *CEP128*[−/−] cells precisely aligned with the distal layer in WT cells, implying a role of distal-layered ODF2 associated with DAP structure (Fig. 7h, i). Histogram analysis verified this result, except that the ODF2-C/N slightly moved toward SCLT1 (Fig. 7j). In addition, to explore the spatial relationship between distal-layered ODF2 and MT triplets, we performed two-color Ex-dSTORM imaging of the ODF2-C/Ac-Tub (Fig. 7k). Our findings showed that ODF2-C was localized on the extension line outside the triplets (dashed lines, Fig. 7k), representing group 1 of ODF2-C (Fig. 6l). This finding illustrates that group 2 of ODF2-C refers explicitly to the proximal-layered ODF2-C that supports

the formation of sDAPs. By contrast, the images of the ODF2-N/Ac-Tub exhibit that the ODF2-N are distributed at the periphery of MT triplets (Fig. 7l). Together, we have constructed the composite image of these three proteins to uncover the relationships among ODF2-C, ODF2-N, and MT triplets (Fig. 7m).

### Distal-layer of ODF2 as an auxiliary role for coordinating and maintaining the DAP structure

Although the previous studies indicated that ODF2 was not directly responsible for DAP formation in mammalian cells[10,47–49], our Ex-dSTORM images indeed uncover a high spatial correlation between the distal-layered ODF2 and the DAP structure. Since the previous genetic analyses of changes in DAP in either *siODF2* or *ODF2*[−/−] cells were mainly achieved by traditional fluorescence microscopy[10,47–49], we speculate on whether any subtle difference at a molecular level may be caused due to absence of *ODF2*. Therefore, we further performed the Ex-dSTORM imaging of the DAP or DAP-associated proteins upon *ODF2* depletion (CRISPR/Cas 9 knockout) to identify the possible structural alterations between WT and *ODF2*[−/−] cells (Supplementary Fig. 17). Interestingly, in the *ODF2*[−/−] cells, several proteins show that puncta are missing at some locations of the iconic ninefold symmetric arrangement (Supplementary Fig. 17a, magenta arrowhead indicates the missing puncta). Out of these proteins that displayed at least a circular pattern, FBF1 was the most prominent structure that was grossly impacted and even deemed unidentifiable in the single-color images. Subsequently, intrigued by this discovery, we performed the two-color Ex-dSTORM imaging of the pair FBF1-SCLT1 in *ODF2*[−/−] cells, where SCLT1 was preserved relatively more, to reveal the arrangement of FBF1 and to inspect its structural defects (Supplementary Fig. 17b). We noticed a disordered spatial organization of FBF1 proteins, puncta exhibiting at the periphery as well as the center near centriolar triplets. Hence, based on these observations, we conceived that the number of disorganized DAP structures is considerably affected by *ODF2* depletion. In addition to FBF1, we also noticed that additional signals were displayed at the inner of the CEP164 ring, which was not discovered in WT but in *ODF2*[−/−] cells. On the other hand, it is interesting to point out that MNR and CEP90, whose radius is smaller than ODF2, seem not to be affected on their arrangement by *ODF2* depletion. Nonetheless, C2CD3, located at the centriolar lumen with an even smaller radius, is found to be frequently lost in one of nine puncta (already checked with saturated image). Last but not least, as the ninefold symmetry of CEP89 is held in *ODF2*[−/−] cells, its charity becomes more evident, which may result from the absence of the proximal layer of CEP89 owing to the sDAP depletion by *ODF2* KO (CEP89, Supplementary Fig. 17a).

To perform quantitative analysis, we first calculated the percentage of the non-ninefold symmetric pattern in nine selected proteins upon *ODF2* depletion (Supplementary Fig. 18a). Notably, over 80% of FBF1 present non-ninefold symmetric distribution in *ODF2*[−/−] centrioles. Moreover, C2CD3, CEP89, SCLT1, and CEP164 demonstrate over one-third of irregular patterns in *ODF2*[−/−] cells. To confirm that this phenomenon is mainly found in *ODF2*[−/−] cells, we compared those proteins exhibiting over one-third of irregular patterns in *ODF2*[−/−] cells to that in WT cells. Surprisingly, all five proteins display considerable differences, and FBF1 shows the most significant difference of all ($P < 0.0001$) (Supplementary Fig. 18b). Besides inspecting the structural organization, we also compared the ciliation rate (24 h serum withdrawal) between WT and *ODF2*[−/−] cells. Conspicuously, the data indicate that the ciliation in *ODF2*[−/−] cells is much lower than that in WT cells (Supplementary Fig. 18c). These results confirm that ODF2 is not required for the presence of DAPs. However, ODF2 plays a role in maintaining the exact ninefold symmetry of the DAP structure. Loss of ODF2 may lead to the irregular localization of DAP structure, especially for FBF1, thought of as a gating protein in the DAPs, and thus further affects the ciliogenesis. As a result, our finding suggests that the distal layer of

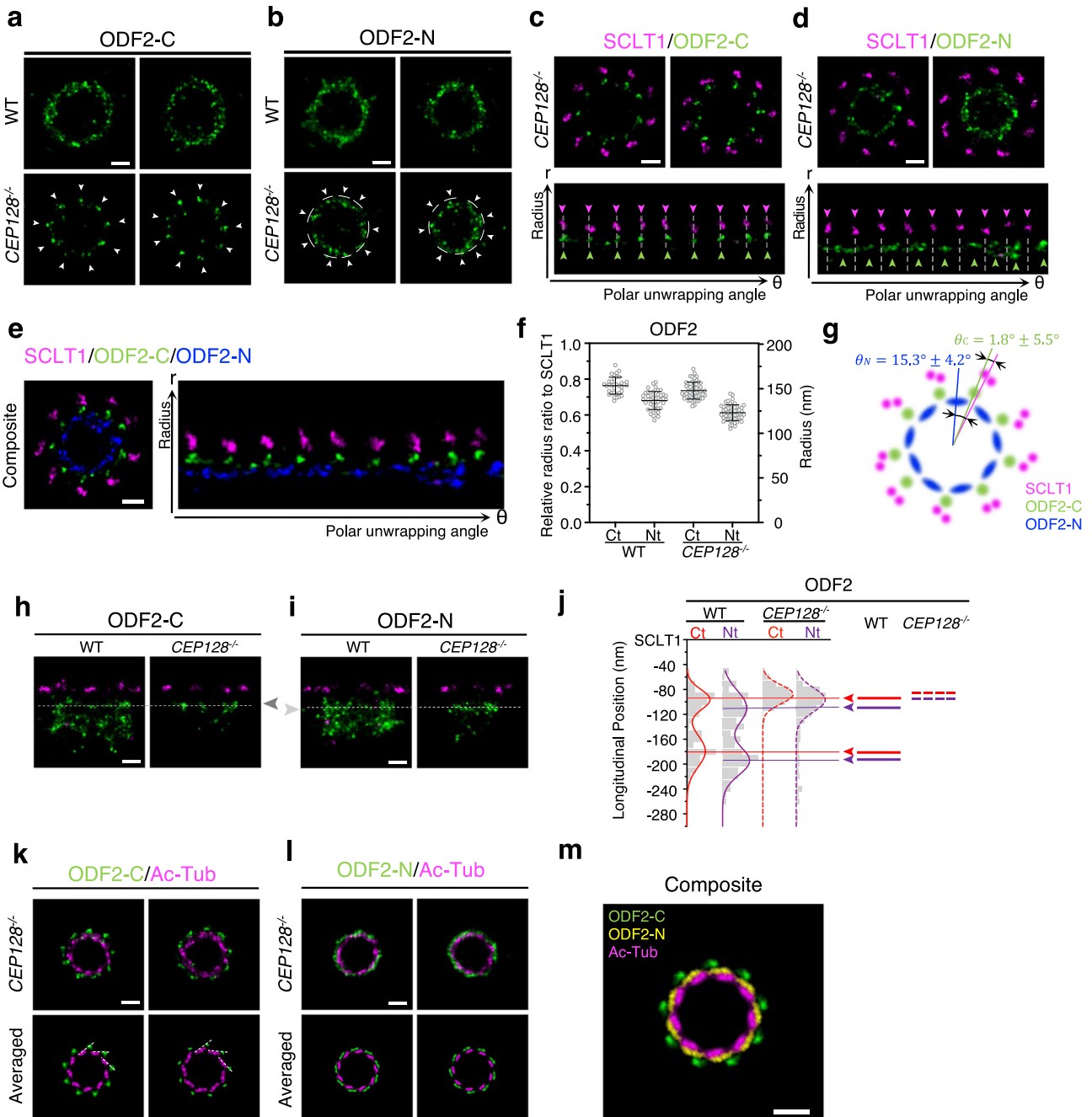

**Fig. 7 | Two-color Ex-dSTORM unraveling the spatial relationship among the distal-layered ODF2, SCLT1, and centriole. a, b** Representative Ex-dSTORM axial-view images of the C-terminus (**a**) and the N-terminus (**b**) of ODF2 in WT and *CEP128⁻/⁻* cells. The well-resolved ninefold radial symmetry re-emerged in sDAP-depleted (*CEP128⁻/⁻*) cells (arrows). **c, d** Two-color Ex-dSTORM images allowing for differentiation of distinct patterns of ODF2-C/N with SCLT1 in *CEP128⁻/⁻* cells. Polar unwrapping analysis in (**c**) reveals that ODF2-C aligns well with SCLT1, while ODF2-N in (**d**) locates between two SCLT1 puncta. **e** Composite images of (**c**) and (**d**) aligned with SCLT1, showing the interlacing spatial relationship between ODF2-C and ODF2-N. **f** Mean radius analysis revealing differences of ODF2-C/N between WT and *CEP128⁻/⁻* cells (mean ± SD, *n* = 35, 44, 54 and 54 blades for ODF2-C (WT), ODF2-N (WT), ODF2-C (*CEP128⁻/⁻*) and ODF2-N (*CEP128⁻/⁻*), respectively). **g** Schematic model of ODF2-C/N and SCLT1 with quantitative analysis in angle (mean ± SD, *n* = 5 and 6 centrioles for ODF2-C (CEP*128⁻/⁻*) and ODF2-N (*CEP128⁻/⁻*), respetively).

**h, i** Representative Ex-dSTORM lateral-view images of the C-terminus (**h**) and the N-terminus (**i**) of ODF2 in WT and *CEP128⁻/⁻* cells. The remaining layer of ODF2 in *CEP128⁻/⁻* cells presents as the distal layer in WT cells (dashed line). **j** Longitudinal distribution histogram describing ODF2-C/N relative to SCLT1 for WT and *CEP128⁻/⁻* cells. In both cells, ODF2-C is slightly closer to SCLT1. Upon sDAP depletion, ODF2 is seen moving toward SCLT1 (*n* = 6, 5, 6, and 6 centrioles for ODF2-C (WT), ODF2-N (WT), ODF2-C (*CEP128⁻/⁻*) and ODF2-N (*CEP128⁻/⁻*), respectively). **k, l** ODF2/Ac-Tub Ex-dSTORM images in *CEP128⁻/⁻* cells. Averaged images highlight the significantly different arrangement ODF2-C locates on the extension line of Ac-Tub (dashed lines) (**k**) while ODF2-N distributes at the periphery of MT triplets (**l**). **m** Composite image of one representative averaged image from (**k**, **l**) revealing a unique architecture formed by ODF2-C, ODF2-N, and Ac-Tub. Each averaged image is obtained based on the processing described in Supplementary Fig. 7. Scale bars, 100 nm (**a–e**, **h**, **i**, **k–m**). Source data are provided as a Source Data file.

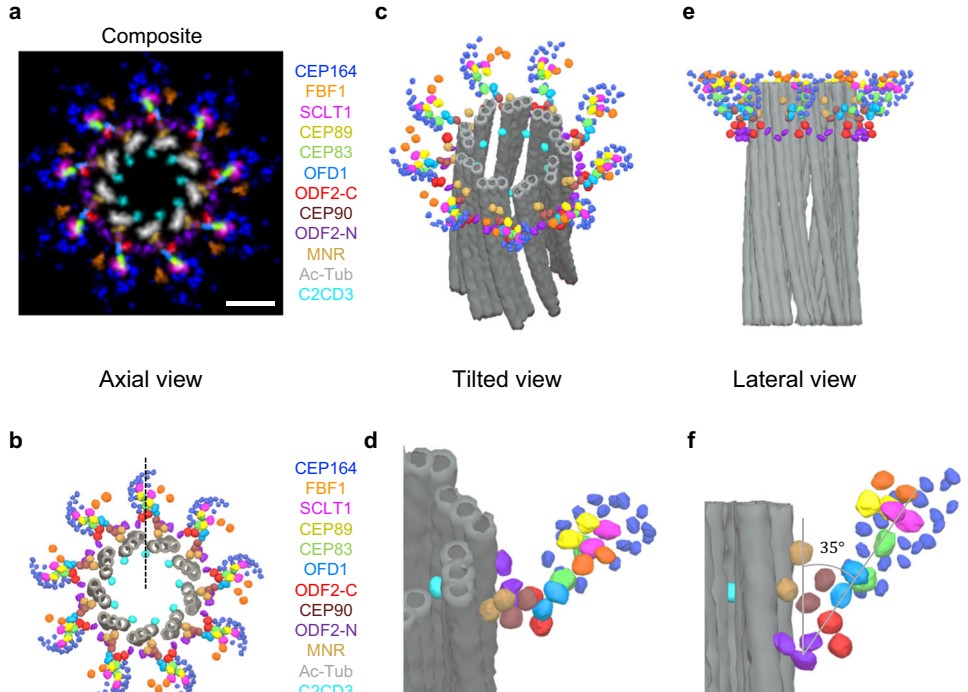

**Fig. 8 | 3D model of DAPs displaying the ultrastructural framework from the centriole wall. a** Composite of rotationally averaged images of the DAP proteins (from Fig. 2e) with Ac-Tub (from Supplementary Fig. 12) and one representative averaged image of MNR, CEP90, OFD1, and ODF2-C/N with scaled dimensions relative to SCLT1. MNR, CEP90, OFD1, and CEP83 jointly construct a linear backbone with anticlockwise chirality against centriole. **b** Axial-view 3D model positioning individual localizations of DAP proteins, MNR, CEP90, OFD1, distal-layered ODF2-C/N, and MT triplets (viewed from the distal end of the centriole). The model reveals the same radial direction among C2CD3, ODF2-C, CEP83, CEP89, and SCLT1 (dashed line). The overall configuration of those proteins manifests a pinwheel-like morphology revealed by the EM study[53] **c, d** A tilted view of the 3D model illustrating the molecular configuration of all proteins against the mother centriole.

Compared with ODF2-C, ODF2-N surrounds MT triplets with a broader distribution along the circumferential direction. A magnified view (**d**) of a single blade manifests a stereoscopic, feather-shaped arrangement of CEP164, which uniquely embraces CEP83, CEP89, and SCLT1. **e** A lateral view of the 3D model revealing the relative longitudinal positions of all proteins against the mother centriole. ODF2-C/N appear at the pedestal of the DAP. **f** An enlarged view of a single blade in (**e**) delineating that a DAP blade is initiated from the centriole wall, scaffolded via MNR, CEP90, and OFD1, and extended to outer DAP components. The distal-layered ODF2 may play an auxiliary role in maintaining the DAP architecture. Quantitative measurement of the protrusion angle is highly in accord with previous EM analyses[26,29]. Scale bar, 100 nm (**a**).

ODF2 plays an auxiliary role in coordinating and maintaining the DAP structure.

## Discussion

In summary, by utilizing optimized two-color Ex-dSTORM, we have demonstrated the power of protein mapping with a resolution of a few nanometers in unraveling the molecular organization of centriolar DAP, vital for functional and ultrastructural studies. In this work, we proposed the in situ drift correction protocol allowing us to observe the organelles inside the expanded cells through dSTORM. Systematical two-color Ex-dSTORM imaging from the axial and lateral views enables us to deduce detailed 3D molecular arrangements of DAP proteins. The direct relationship between DAPs and MT triplets can be discovered at a few-nanometer resolution. Furthermore, we have identified the molecular constitution of the proteins near the base of DAP, including MNR, CEP90, OFD1, C2CD3, and the distal-layered ODF2. Next, taking all the Ex-dSTORM images together, the ultrastructural constitution from the inner wall of MT triplets to the outer DAP is revealed (Fig. 8a). Interestingly, MNR, CEP90, OFD1, and CEP83 cooperatively construct a linear backbone of DAP with anticlockwise chirality. Integrating the information from the various features (chirality, inclination angle, characteristic length, puncta count, mean position) of each DAP protein (Supplementary Tables 2–4), we build a 3D computational model for seeing the single-blade ultrastructure (Fig. 8b–f). Our results manifest the nearly same radial direction among C2CD3, ODF2-C, CEP83, CEP89, and SCLT1 in the 3D model

(dashed line, Fig. 8b). Together with the genetic relationship of C2CD3 as an upstream protein in DAP assembly[18,20,30,31,50], our model shows that C2CD3 near the A-tubule of an MT triplet is involved in the configuration of the DAP assembly. Presumably, DAP blades are developed through a directional correlation of the following proteins: C2CD3, MT triplets, MNR, CEP90, OFD1, CEP83, CEP89, SCLT1, and CEP164. The distinct angular position of FBF1 implies that it may not be a part of the backbone of the DAP blade, and it substantiates the statement that FBF1 performs an essential role in ciliary gating[26]. The 3D model depicts the stereoscopic arrangement of a feather-shaped CEP164 embracing CEP83, CEP89, and SCLT1 (Fig. 8c, d). As for ODF2, our image-based analysis with the molecular resolution has disclosed the previously undetectable alteration from some DAP components observed in *ODF2*-depleted cells.

Furthermore, ODF2-N structurally encircles MT triplets with a broader distribution along the circumferential direction. Therefore, this arrangement may provide more contact sites and have higher mechanical stability, reflecting its importance in maintaining the DAP integrity. In the lateral view, ODF2-C and ODF2-N may jointly construct the pedestal of the DAP proteins (Fig. 8e). A single blade of the 3D model delineates the molecular constitution of DAP from the centriole wall, manifesting our optics-based ultrastructural protein mapping (Fig. 8f).

To perform the Ex-dSTORM imaging of a specific organelle, one should first choose a preferred scheme in sample expansion. Although this work has demonstrated a successful application to the study of

distal appendages, further procedure compatible with other cellular structures remains to be optimized. Second, we have developed the in situ drift correction to enable cell imaging deep into a gel and facilitate cell search in various cellular structures. This implementation offers a robust and straightforward solution in most localization microscopy systems.

On the other hand, the crosstalk from the far-red to red channel has posed severe image artifacts in two-color SMLM, which may stem from the photoblueing effect suggested by a recent study[51]. Nonetheless, our result, AF647 causing higher crosstalk than ATTO647N, seems not to agree with the previous finding, even in the oxygen-scavenging system. One possible explanation is that although ATTO647N possesses higher photoblueing conversion efficiency, its converted state might present a much lower survival fraction of single-molecule blinking compared to that of AF647 at a high intensity of 561 nm illumination. The conclusion in Fig. 2 was drawn based on considerably different protein numbers- C2CD3 and Ac-tub. In reality, with a proper labeling strategy in far-red and red channels (for example, more proteins for the red channel; fewer proteins for far-red), one can still obtain two-color Ex-dSTORM images with negligible crosstalk (Supplementary Fig. 16).

The emerging technique of synergistically combining expansion microscopy and single-molecule localization microscopy sheds light on studying protein complexes at the molecular level. However, practical localization microscopy imaging of gel-specimen composites is still arduous. This research thus proposes an attainable Ex-dSTORM workflow through sample preparation, in situ drift correction, optimized two-color imaging, and ultrastructural image analysis. Together, our Ex-dSTORM analyses provide a pragmatic roadmap to successfully determine the protein organization from the inner MT triplets to the outer DAP in mammalian cells with a molecular resolution, offering an unprecedentedly ultra-detailed framework of DAPs.

## Methods

### Reagents
Sodium acrylate (SA, 97%, 408220, Sigma-Aldrich), Acrylamide (AA, 40%, A4058, Sigma-Aldrich), Acrylamide/Bis-acrylamide (30%, 29:1, 1610156, Bio-Rad), N,N,N',N'- Tetramethylethylenediamine (TEMED, 1610801, Bio-Rad), Ammonium persulfate (APS, 1610700, Bio-Rad), Paraformaldehyde (FA, 16%, 15710, Electron Microscopy Sciences), Sodium dodecyl sulfate (SDS, 0227, VWR Life science), Sodium chloride (NaCl, 31434, Sigma-Aldrich), Tris (1.5 M, pH 8.8, J831, VWR Life science), Bovine serum albumin (BSA, A9647, Sigma-Aldrich), Tween 20 (P137, Sigma-Aldrich), Methyl alcohol (methanol, 15306121, Macron), Phosphate buffered saline (10X PBS, 70011044, Gibco), Dimethyl sulfoxide (DMSO, D8418, Sigma-Aldrich), Bind-silane (abx082155, Abbexa), Acetic acid (33209, Sigma-Aldrich), Ethanol (absolute, ≥99.8%, 32221, Sigma-Aldrich).

### Cell culture
Human retinal pigment epithelial cells (hTERT RPE-1, ATCC-CRL-4000) were cultured in Dulbecco's modified Eagle's medium (DMEM)/F-12 mixture medium with L-glutamine and HEPES (1:1; 11330-032, Gibco, Thermo Fisher Scientific) at 37 °C under 5% CO$_2$ with 10% fetal bovine serum (FBS, SH3010903, Hyclone), sodium bicarbonate (NaHCO$_3$, S6014, Sigma-Aldrich), and 1% penicillin-streptomycin. Prior to fixation, RPE-1 cells were cultured on poly-L-lysine coated coverslips, and cilium formation was induced by 24–48 h serum starvation. Then the cells were fixed with cold methanol at −20 °C for 10 min.

### Antibodies
Detailed information on the primary antibodies used in this work is listed in Supplementary Table 5. The second antibodies used in this study were Alexa Fluor 647 (dilution 1/100, anti-mouse A31571, anti-rabbit A31573, anti-rat A21247, anti-goat A-21447; Thermo Fisher

Scientific), Atto 647 N (dilution 1/100, anti-mouse 50185-1ML-F; Sigma-Aldrich), CF568 (dilution 1/100, anti-rabbit 20098, anti-rat 20092, anti-mouse 20105; Biotium), and Alexa Fluor 488 (dilution 1/100, anti-mouse A21202; Sigma-Aldrich). Dyomics 654 (Dy654)-conjugated secondary antibodies were custom-made by conjugating Dy654 N-hydroxysuccinimidyl (NHS) ester (654-01; Dyomics) to different IgG antibodies respectively (anti-mouse 715-005-151, anti-rabbit 711-005-152, anti-rat 712-005-153; Jackson ImmunoResearch).

### CRISPR construction of *CEP128*$^{-/-}$ and *ODF2*$^{-/-}$ cells
In order to inactivate CEP128 and ODF2 in RPE-1 cells, the RNA-guided targeting of genes was achieved through coexpression of the Cas9 protein with gRNAs using reagents from the Church group[52] (Addgene: http://www.addgene.org/crispr/church/). Targeting sequences of the gRNAs were as followed: *CEP128* gRNA2 (5'-GCTGCCAGATCAACGCA-CAGGG-3'), CEP128 gRNA4 (5'-GAGTCAGCTCTGAGATCTGAAGG-3'), *CEP128* gRNA5 (5'-GCAGCTGAACTTCAGCGCAATGG-3'), *ODF2* gRNA1 (5'-GAGGGAACAGCACTGCAAAGAGG-3'), *ODF2* gRNA2 (5'-GAGT GTCCGGGTGAAAACCAAGG-3'). To achieve complete protein depletion, we adopted multiple gRNAs targeting different exons. These targeting sequences were cloned into the gRNA vector (plasmid 41824, Addgene) via the Gibson assembly method (New England Biolabs). Pure *CEP128* and *ODF2* knockout cells were obtained through clonal propagation from a single cell and then confirmed by genotyping and immunoblotting as in previous researches[25,47].

### Sample expansion and immunofluorescence
RPE-1 cells on 12 mm coverslips were fixed with methanol at −20 °C for 10 min. Cells were then incubated in the perfusion solution containing 1.4% FA and 2% AA in PBS for 5–6 h at 37 °C. Next, gelation was proceeded via incubating each coverslip with U-ExM monomer solution (19% (w/w) SA, 10% (w/w) AA, 0.1% (w/w) BIS in PBS) supplemented with 0.5% TEMED and 0.5% APS in order in a pre-cooled humidified chamber. After pre-polymerization on the ice for 1 min, the chamber was transferred into the incubator for gelation at 37 °C for one hour. Coverslips with hydrogel were placed in fresh denaturation buffer (200 mM SDS, 200 mM NaCl in 50 mM Tris (pH8.8)) for 15 min at RT with gentle shaking. Detached hydrogels were incubated in fresh denaturation buffer at 95 °C for 1.5 h. After denaturation, the hydrogels were expanded in fresh double deionized water (ddH$_2$O) at least three times (30 min for each) with gentle shaking and then incubated overnight in ddH$_2$O. Next, the hydrogels were kept in PBS before immunostaining. Immunostaining was first carried out via labeling the trimmed hydrogels with primary antibodies diluted in 2% BSA/PBS at 37 °C in the 1.5 mL Eppendorf for three hours with gentle shaking, followed by washing three times with 0.1% Tween 20 in PBS and twice with PBS for 20 min each. Next, the hydrogels were stained with secondary antibodies diluted in 2% BSA/PBS at 37 °C for three hours with gentle shaking followed by the same washing steps. Finally, the hydrogels were expanded in the ddH$_2$O until reaching their maximal expansion via exchanging ddH$_2$O at least three times.

### Re-embedding of expanded hydrogels
A neutral acrylamide gel was crosslinked across the expanded hydrogel for stabilization in the dSTORM imaging buffer. The expanded hydrogels were incubated in freshly prepared re-embedding gel solution (10% (w/w) AA, 0.15% (w/w) BIS, 0.05% (w/w) TEMED, 0.05% (w/w) APS in ddH$_2$O) twice for 25 min each time at RT with gentle shaking. The polymerization process was facilitated by sandwiching the hydrogel with two untreated coverslips in a nitrogen-filled humidified chamber and incubating at 37 °C for 1.5 h. After polymerization, the re-embedding gels were washed three times in ddH$_2$O for 20 min each. The re-embedded hydrogels were mounted in our customized chamber prior to imaging. On the other hand, to compare the expansion factor at different gelation and re-embedding conditions

(Supplementary Fig. 2), the experiment using bind-silane coating was conducted for free radical polymerization by incorporating the acryloyl groups onto the coverslips on which the re-embedded gels were immobilized. That is, the uncharged acrylamide gel was crosslinked through the polymerization process across the hydrogel and the treated coverslips to facilitate the chemical binding of hydrogel onto the coverslip. We first prepared the fresh working solution containing 5 µL bind-silane, 8 mL absolute ethanol, 200 µL acetic acid, and 1.8 mL ddH$_2$O. Then coverslips were washed with ddH$_2$O and absolute ethanol and rinsed with the working solution. After the working solution was fully evaporated, the coverslips were washed with absolute ethanol and then allowed to be air-dried. Then, the expanded hydrogel incubated in the re-embedding solution was transferred onto the bind-silane-treated coverslips. The following polymerization process was the same as described above.

### Determination of expansion factor

In addition to quantifying the ratio of the gel size before and after expansion, SCLT1 was further utilized as the structural reference to determine the microscopic expansion factor after re-embedding throughout this study. A nearly perfect top view of SCLT1 was imaged via dSTROM and Ex-dSTORM, respectively. The expansion factor is then determined by the ratio of the mean diameter of SCLT1 between Ex-dSTORM and dSTORM.

### Ex-dSTORM imaging

Ex-dSTORM image acquisition was performed on a custom-built setup based on a commercial inverted microscope (Eclipse Ti-E, Nikon) with a focus stabilizing system and a laser merge module (ILE, Spectral Applied Research) with individual controllers for four light sources. To perform the dSTORM imaging across the expanded cells, we adopted epi-illumination in our optical system. For wide-field illumination of samples, beams from a 637 nm laser (OBIS 637 LX 140 mW, Coherent), a 561 nm laser (Jive 561 150 mW, Cobolt), a 488 nm laser (OPSL 488 LX 150 mW, Coherent), and a 405 nm laser (OBIS 405 LX 100 mW, Coherent) were homogenized (Borealis Conditioning Unit, Spectral Applied Research) and focused onto the back focal plane of an oil-immersing objective (100 × 1.49, CFI Apo TIRF, Nikon). During Ex-dSTORM image acquisition, the 637 nm and 561 nm laser lines were operated at a high intensity of ∼3 kW/cm$^2$ to quench most of the fluorescence from Alexa Fluor 647, Dy654, and CF568. A weak 405 nm laser beam was introduced to convert a portion of fluorophores from a dark to a fluorescent state. The 488 nm laser line was intermittently switched on every 800 frames for in situ drift correction. The fluorescent signal was filtered through a quad-band filter (ZET405/488/561/640 mv2, Chroma). Then the signal was recorded on an electron-multiplying charge-coupled device (EMCCD) camera (Evolve 512 Delta, Photometrics) with a pixel size of 93 nm. For single-color imaging, signals of Alexa Fluor 647 were filtered with a quad-band filter; for two-color imaging, the Dy654 channel was first obtained, then the CF568 channel was acquired with a combination of quad-band and short-pass filter (BSP01-633R-25, Semrock). Generally, for each dSTORM image, 15,000–30,000 frames were acquired at a rate of 50 fps. The position of the individual single-molecule peak was then localized using Meta-Morph Superresolution Module (Molecular Devices) based on a wavelet segmentation algorithm. The superresolution images were cleaned with the gaussian filter of 0.7–1 pixel.

The re-embedded hydrogels were immobilized within a customized holder and incubated in an imaging buffer containing Tris-HCl, NaCl (TN) buffer at pH 8.0, 10–100 mM mercaptoethylamine (MEA) at pH 8.0, and an oxygen-scavenging system consisting of 10% glucose (G5767, Sigma-Aldrich), 0.5 mg mL$^{-1}$ glucose oxidase, and 40 µg mL$^{-1}$ catalase.

To validate the system resolution of our dSTORM imaging, the Fourier ring correlation (FRC) analysis was performed for the experimental image using an ImageJ plugin (GDSC SMLM). The shortest distance between two labeled molecules that can be distinguished with Ex-dSTORM was estimated by further dividing the dSTORM resolution by the average expansion factor (Supplementary Fig. 5).

### In situ drift correction

To perform in situ drift correction, marker protein (ATP synthase in this study, ab109867, Abcam) was first labeled with Alexa Fluor 488 at a dilution of 1/100 before imaging. Lateral position drift was intermittently measured during acquisition and corrected by an ImageJ plugin (details in Supplementary Note 2 and Supplementary Fig. 3). The sets of images with maker signals were eliminated with a home-made code based on Labview, Matlab, and ImageJ before localizing each single-molecule peak. Chromatic aberration between red and far-red channels was compensated with a customized algorithm relocating each pixel of a 561 nm image to its targeted position in the 647 nm channel with a predefined correction function obtained by a parabolic mapping of multiple calibration beads.

### Imaging analysis

For axial imaging, a ring pattern of proteins in this research was used to identify the top view orientation. To determine the diameter of these proteins with minimal variations caused from batches to batches, cells to cells, or blades to blades of the sample, we used SCLT1 as the reference to measure the radius ratio of proteins (Supplementary Fig. 11) and scaled accordingly. For dispersed radial distribution of CEP164, radial positions were described with the radius ratio of each punctum with respect to SCLT1. To determine the relative angular position among DAP proteins, we estimated angle differences of puncta between the proteins and SCLT1. More details are illustrated in Supplementary Fig. 8. For lateral-view imaging, a rod-like pattern was used to ensure the orientation. To determine the relative longitudinal position among DAP proteins, we quantified relative longitudinal distances between each punctum of proteins and SCLT1 (Supplementary Fig. 11). All the images of expanded samples were scaled by the expansion factor of 3.92. The angular correlation in Fig. 6b was determined based on the maximum correlation coefficient between C2CD3 and Ac-Tub signals. The same procedure was performed in Fig. 7g of the ODF2-N/SCLT1 protein pair due to its wider distribution in the circumferential direction.

### Three-dimensional model illustration

The 3D model of the DAP was constructed with the 3D illustration software Blender (Blender Foundation). The dimensions of the DAP model were based on the mean protein positions obtained from Ex-dSTORM analyses in the results (Supplementary Tables 2–4). The ultra-detailed information of DAP molecular localizations from each protein was then applied to the DAP model correspondingly. The 3D model of the centriolar microtubules was based on the information from the previous study[53].

### Statistics and reproducibility

The statistical significance of differences among groups was evaluated with a two-tailed, unpaired Student's $t$-test using GraphPad Prism. Differences were considered statistically significant at $p < 0.05$. The statistical test used, and the significance are always reported in the figure legends. All Ex-dSTORM images (Figs. 2a, 4a–e, 5a, 6a, d, e, h–j, l, m, 7a–d, h, i, k–l; Supplementary Figs. 3a–c, 5a, 6, 15a, 16, 17a, b) panels were repeated at least three times and representative images were shown in this paper. All data points for analytical results and statistical analyses are provided in the Source Data file.

### Reporting summary

Further information on research design is available in the Nature Portfolio Reporting Summary linked to this article.

## Data availability

All the data supporting the findings described in this study are available within the article, Supplementary Materials. The analytical data generated in this study are provided in the Supplementary Information. Source Data are provided with this paper.

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

## Acknowledgements

We thank Jung-Chi Liao and Meng-Fu Bryan Tsou for sharing reagents and cell lines. This work was supported by the National Science and Technology Council, Taiwan (Grant No. 109-2222-E002-003-MY3 and Grant No. 111-2628-E-002-021-) to T.T.Y. and the "National Taiwan University Higher Education Sprout Project (NTU-111L8809)" within the framework of the Higher Education Sprout Project by the Ministry of Education (MOE) in Taiwan.

## Author contributions

T.T.Y. and T.-J.B.C. conceived the study. T.-J.B.C. and J.C.H performed the Ex-dSTORM imaging and analyzed the data. T.T.Y. provided guidance and assistance on the project. T.-J.B.C. and T.T.Y. created the 3D computational model. T.-J.B.C., J.C.H., and T.T.Y. interpreted the data and wrote the manuscript.

## Competing interests

The authors declare no competing interests.
