## [Peer Review File · Nature Communications]

REVIEWER COMMENTS

Reviewer #1 (Remarks to the Author):

In this manuscript entitled "Single-molecule localization microscopy reveals the ultrastructural root constitution of distal appendages in expanded mammalian centrioles" Chang and colleagues use 2-colors direct stochastic optical reconstruction microscopy (dSTORM) on expanded centrioles to decipher distal appendage proteins (DAP) localization with a resolution of approximately 5 nm. The authors develop a new strategy for their image acquisition incorporating an in-situ drift correction. In addition, they optimize the combination of dyes to yield minimum spectral cross-talk. As a result, the molecular organization of the major distal appendage proteins is resolved. Therefore, this study refines

the previous work of Yang et al (2018), Bowler et al (2019) and Chong et al (2020) allowing a precise mapping of the major DAP (CEP83, CEP89, SCLT1, CEP164) together with C2CD3 protein, required for distal appendage assembly and ODF2 protein, a protein belonging to both distal and subdistal appendages, with respect to the microtubule wall.

In its present form, the paper appears as a mix between a technical paper and a scientific paper.

However, as a technical paper, two papers have already been published using similar techniques (1 and

2 colours Ex-STORM; Xu et al, 2019 and Zwettler et al, 2020). As a scientific paper, despite the images are of good quality and nicely analysed, the resulting data provide only a more precise view of the molecular organization of distal appendages that was already known.

I have some major concerns:

1) Concerning the technique used, the authors used an in-situ drift correction. It is nicely explained how they proceed using a drawing Fig 1c, but no Figures showing the differences in acquisition before the drift correction and after the drift correction are provided. The authors re-embed the expanded gel in ddH₂O rather than Tris buffer. The authors could provide a direct comparison of the two methods by showing the data and their analysis, or they could decide to precise these differences only in mat and meth. The authors claim having a 3nm resolution, while in the two other publications using similar techniques the resolution is around 10nm. This difference should be explicitly explained and discussed. Finally, it is not clear for the reader what was already known and what is new. The publication of Zwettler et al, 2020 should be cited, for example line 117, since the authors use part of the published protocol. In addition, the Zwettler publication applies the ExdSTORM technic to microtubules and Chlamydomonas centrioles.

2) The authors claim that C2CD3 labeling displays a 9-fold symmetry in the lumen of the centriole wall. This result is partly consistent with the authors' own data previously published (Yang et al 2018), since in this paper the authors reports that C2CD3 localize compactly in the centriolar lumen as observed by STORM microscopy. Recently, Figure 6 of the Gaudin and colleagues (2022) shows an asymmetric labeling of C2CD3 in the centriole lumen on expanded centriole. How do the authors reconcile this discrepancy? In any case, the Gaudin paper should be cited.

3) In the title of the paper, the authors mention an "ultrastructural root constitution of distal appendages". It is not clear to me what the authors are thinking by "root". I interpret that "root" means the requirement of the root protein to drive distal appendage assembly. Therefore, the delocalization of the root protein should recruit appendage proteins. In this case the experiment should be performed.

4) It is intriguing that the authors define ODF2 as the root protein, since the depletion of ODF2 in RPE1 (Tanos et al, 2013; Kuhns et al 2013 and Viol et al 2020) does not affect the presence of distal appendages as observed by electron microscopy. If my interpretation of root is correct, I would suggest that the authors deplete ODDF2 and look at the assembly of distal appendages with their own cell line (RPE1) which might be different from the other teams.

5) Two recent papers: Kumar et al 2021 (Journal of cell biology) and LeBorgne et al, 2021 (BioRxiv 2021.07.13.452210) show that the complex containing MNR, CEP90, OFD1 and FOPNL were required for distal appendage assembly. In these two papers, the localization of the different member of this complex has been analyzed using pre- and post- expansion labeling. The results show that these proteins are located between γ -tubulin or α and β tubulin used as a reference for the centriolar microtubule wall and the most-proximal distal

appendage protein CEP83. Surprisingly the authors do not cite these papers.

6) Line 163, the authors claim that C2CD3 is a core distal appendage protein. C2CD3 is required for distal appendage assembly, however its localization in the centriolar lumen prevents this hypothesis as already suggested by the corresponding author of this paper (Yang et al, 2018).

7) Line 330, The gap between CEP83 and Ac-tub was already observed using post-labeling expansion microscopy in Gaudin et al, 2022 and Le Borgne et al, BioRxiv, 2021.

8) Line 363, "our finding suggests that OFD2 fills the undefined coverage, ...". This statement excludes other proteins already known (such as MNR, FOPNL, CEP90 and OFD1) or to be discovered that localize there.

9) Line 388, the authors claim that "the morphology of DAP would alter upon sDAP depletion". In their previous paper (Chong et al, 2020), the same conclusion was already reached. I would have expected a comment such as "as previously shown by ...":

Other comments

10) Line 445-447: The authors talk about a 18° offset between the 2 layers of OFD2. Can the authors explain to which data they refer?

11) Some scale bars are missing.

12) The representation of the measured angle could have been done in an easier way.

13) Material and methods could have been better explained as well as figure legends

Reviewer #2 (Remarks to the Author):

The manuscript used Ex-dSTORM to study the molecular organization of centrioles with a particular focus on the root structure of centriolar DAP in cells. After ~4-fold expansion in charged polyacrylamide gel the sample was re-embedded to avoid shrinking of the gel in photoswitching buffer. Using an optimized re-embedding protocol the authors achieved a final expansion factor of 3.9, i.e. slightly larger than shown in a previous Ex-dSTORM paper. While previous works have shown 3D Ex-dSTORM with a single color, the present work used two-color 2D to map the position of different immunolabeled proteins. Performing frontal and side view 2D dSTORM images the authors could finally reveal detailed information about the molecular arrangement of DAP proteins. I recommend acceptance of the manuscript in Nat Commun but request the authors to tone down a few statements that are as presented untenable.

A previous Ex-dSTORM study of centrioles resolved at least partially the microtubule triplets as hollow structures (Ref. 41). In the present manuscript the tubulin signal (e.g. Fig. 2) indicates a substantially lower spatial resolution, why? Can the authors comment of that?

In the introduction the authors state that they will introduce a method that enables Ex-dSTORM imaging throughout the cell but then they use TIRF illumination with an oil-immersion objective. This needs more explanation.

Even though the authors use TIRF they achieve only a localization precision of ~12 nm, i.e. lower than in other dSTORM studies. This corresponds to a spatial resolution of ~30 nm. Hence, with 4-fold expansion the method can achieve ~8 nm spatial resolution (best case scenario). But, a spatial resolution of ~10 nm is still far away from a resolution well close to an EM level as claimed in the intro. I request the authors to perform FRC on their images or another resolution estimation to come up with realistic resolution claims.

Furthermore, the localization precision remains at 12 nm even in the expanded sample. It is wrong to divide the loc precision with the expansion factor and then state 3 nm loc precision.

Finally, I do not understand why the authors tested ATTO647N for dSTORM imaging since it is known that ATTO47N does not show photoswitching in thiol-buffer. In addition, it has been shown that ATTO647N shows dramatic photobleaching (Ref. 46).

Reviewer #1 (Remarks to the Author):

In this manuscript entitled “Single-molecule localization microscopy reveals the ultrastructural root constitution of distal appendages in expanded mammalian centrioles” Chang and colleagues use 2-colors direct stochastic optical reconstruction microscopy (dSTORM) on expanded centrioles to decipher distal appendage proteins (DAP) localization with a resolution of approximately 5 nm. The authors develop a new strategy for their image acquisition incorporating an in-situ drift correction. In addition, they optimize the combination of dyes to yield minimum spectral cross-talk. As a result, the molecular organization of the major distal appendage proteins is resolved. Therefore, this study refines the previous work of Yang et al (2018), Bowler et al (2019) and Chong et al (2020) allowing a precise mapping of the major DAP (CEP83, CEP89, SCLT1, CEP164) together with C2CD3 protein, required for distal appendage assembly and ODF2 protein, a protein belonging to both distal and subdistal appendages, with respect to the microtubule wall. In its present form, the paper appears as a mix between a technical paper and a scientific paper. However, as a technical paper, two papers have already been published using similar techniques (1 and 2 colours Ex-STORM; Xu et al, 2019 and Zwettler et al, 2020). As a scientific paper, despite the images are of good quality and nicely analysed, the resulting data provide only a more precise view of the molecular organization of distal appendages that was already known.

Reply: This paper is indeed a mix between a technical and a scientific paper, and we will specify our major contributions below. For the technical part, we address the challenge of generally applying Ex-dSTORM for biological research. Owing to the extended distance between the cellular structure of interest and the coverslip, performing drift correction using fiducial markers on coverslips becomes arduous in the Ex-dSTORM imaging. Therefore, we speculated that this might be why the published papers studied the cellular subjects near the coverslip (murine spermatocyte cell spread, Xu et al., 2019; microtubules and isolated *Chlamydomonas* centrioles, Zwettler et al., 2020). In this work, we first proposed the in-situ drift correction for addressing this issue which was not elaborated in these papers. Second, we found a severe crosstalk phenomenon in the two-color Ex-dSTORM imaging, which may lead to the mislocalization of the proteins. Hence, in this study, we suggested an ideal combination of fluorescent dyes for conducting two-color Ex-dSTORM experiments with high fidelity. Third, we developed an optimized protocol for sample preparation, and the size of specimen-gel composites was nicely retained, giving an expansion factor close to 4, not 3 or 3.4 achieved in previous works. Thus, it further enhances the overall resolution by at least ~ 15 % and reduces the linkage error of antibodies.

In terms of the scientific part, we performed our Ex-dSTORM to manifest the ultrastructural

organizations of several distal-end proteins for the centriole, revealing their chirality and precise molecular arrangement, which were not observed before (**Figure R1**). Moreover, we uncovered the nine-fold symmetric pattern of the C2CD3 protein with high spatial correlation with distal appendage proteins (aligning in the same radial direction). Finally, our result indicated that the distal-layered ODF2 showed a high spatial correlation with the distal appendage proteins. Surprisingly, upon ODF2 depletion, our Ex-dSTORM imaging revealed the disorganized localization (non-ninefold symmetry) for some distal appendage proteins, and thus ciliation was perturbed, suggesting that ODF2 may function as a coordinator for distal appendage integrity. This image-based study elucidated how ODF2 affected the distal appendage at the structural level and clarified its role in the distal appendage, whose assembly was thought to be less related to ODF2 based on previous understanding.

Figure R1 Ex-dSTORM imaging of DAPs manifesting the ultrastructural framework from the centriole wall. Composite of rotationally averaged images of the DAP proteins with Ac-Tub and one example image of MNR, CEP90, OFD1, and ODF2-C/N with scaled dimensions relative to SCLT1. MNR, CEP90, OFD1, and CEP83 jointly construct a linear backbone with anticlockwise chirality against centriole. Scale bar, 100 nm.

I have some major concerns:

1. Concerning the technique used, the authors used an in-situ drift correction. It is nicely explained how they proceed using a drawing Fig 1c, but no Figures showing the differences in acquisition before the drift correction and after the drift correction are provided. The authors re-embed the expanded gel in ddH₂O rather than Tris buffer. The authors could provide a direct comparison of the two methods by showing the data and their analysis, or they could decide to precise these differences only in mat and meth. The authors claim having a 3nm resolution, while in the two other publications using similar techniques the resolution is around 10nm. This

difference should be explicitly explained and discussed. Finally, it is not clear for the reader what was already known and what is new. The publication of Zwettler et al, 2020 should be cited, for example line 117, since the authors use part of the published protocol. In addition, the Zwettler publication applies the ExdSTORM technic to microtubules and Chlamydomonas centrioles.

Reply: We would like to thank the reviewer for the constructive comments. The reviewer has posed a concern that no figures in our studies were provided to manifest the disparities before and after drift correction. To elaborate on this work, we present the practical procedure of in-situ drift correction in **Figure R2**. In **Figure R2a**, we provide the images of both the in-situ marker (ATP synthase stained with AF488) and target protein (Ac-Tub labeled with AF647) in its corresponding channels for clarity. The region of interest (ROI) for correction and Ex-dSTORM imaging were marked respectively for the following processing. The image acquisition process is composed of incessant imaging channel excitation (637- or 561-nm laser) with intermittent 488-nm laser firing (**Figure R2b**). It becomes convenient to discern the on-and-off states of in-situ markers by measuring intensity in correction ROI, as shown in **Figure R2b**, where each burst is due to 488-nm excitation. From here, the lateral drift against time was plotted according to the correlation of in-situ markers across all frames taken in the correction ROI (**Figure R2b**). The imaging series in the target ROI was then corrected and processed with its corresponding in-situ drift correction result. Sample image pair of the corrected and uncorrected condition is shown, and a special remark is given to the ambiguity of molecular structure displayed in the uncorrected image and the enhancement of detail shown in the corrected image (**Figure R2c**). Exceptionally, our target centriolar proteins (Ac-Tub) can be preserved with molecular details where iconic triplets are resolved following in situ drift correction. Although we could identify the nine cluster signals from the uncorrected image, the microtubule triplet arrangement could not be revealed. After in-situ drift correction, we could unravel the arrangements of microtubule triplets as marked with dashed boxes and arrowheads. To clarify this, we have added a description of the in-situ drift correction about **Figure R2** to the supporting information (**Supplementary Note 2** and **Supplementary Figure 3**).

In response to the reviewer's comment on the comparison of gel preparation protocol, we performed an experimental trial to strengthen that gel re-embedded in ddH₂O can retain as much expansion factor as possible, compared with that in Tris buffer. In order to directly compare the differences between our re-embedding process with others, we take two conditions into account – with or without Tris buffer and with or without bind-silane coating. The 2-cm hydrogel was re-embedded with either commonly-used bind-silane coating or without and compared in terms of its length changed in two solutions (**Figure R3**). We first prepared two solutions, and both of these

Figure R2 Workflow of in-situ drift correction. **a** Representative images of in-situ marker and target protein in the correction channel (488-nm laser excitation) and imaging channel (637- or 561-nm laser excitation), respectively. Green and blue boxes represent the regions of interest (ROI) for in-situ drift correction and target imaging range. The in-situ correction will be processed in the imaging channel. Here we provide the image of an in-situ marker for clarity. **b** During the image acquisition process, the in-situ marker (stained with AF488) was intermittently illuminated (for every 800 frames) in the imaging channel. The corresponding images of single-molecule blinking with the on or off state of the in-situ marker were shown. Following the on-and-off switching of in-situ markers, the lateral drift of in-situ markers was obtained by tracking the location of markers, shown in the in-situ drift correction result. **c** Comparison of uncorrected and corrected Ex-dSTORM images. The corrected image is obtained by post-processing position compensation according to the in-situ drift correction result. Scale bars, 500 nm (**a, b**), 100 nm (**c**).

contained 10% acrylamide, 0.15% N,N'-methylenebisacrylamide, 0.05% TEMED, and 0.05% APS; here, the only difference was in the addition of either Tris buffer (in Solution-1, **Figure R3a** and **b**) or ddH₂O (in Solution-2, **Figure R3a** and **b**). Our result indicated that the re-embedding

Figure R3 Comparisons and optimizations of different re-embedding processes. **a-b** Two solutions listed on the top were prepared for re-embedding processes and expanded hydrogels were trimmed to the size of 2 cm in length. The images of the gel were taken following re-embedding and rehydration processes in two solution conditions with or without bind-silane coating, **a** for two re-embedding processes with bind-silane coating and **b** for the processes without the bind-silane coating. **c** Size of the hydrogel in each condition from **a** and **b** was measured and quantitatively analyzed with the introduction of retention rate, defined by the length measured divided by that of expanded hydrogel (2 cm) prior to re-embedding. The blue-labeled result marks the condition of the previous report¹; the red-labeled result indicates the condition in our experiment.

process without Tris buffer could enhance the retention rate by ~3% in one dimension compared with the process with Tris buffer (**Figure R3c**). In the identical process of post-re-embedding to rehydration in PBS, the length of the hydrogel free of bind-silane binding increases by 11% (from 1.66 cm to 1.84 cm), compared to that with bind-silane binding, valid for either re-embedding solutions (**Figure R3c**). By direct comparison of the two solutions as well as the coating methods, we found two factors that make our re-embedded hydrogel achieve a higher expansion factor than the other work^{1,2} by overall enhancement of at least ~14% (from 1.66 cm to 1.89 cm). We have included this result in the supporting information (**Supplementary Figure 2**) and added sentences in the revised manuscript (Line 123 and **Supplementary Note 1**) to clarify this point.

Owing to a higher expansion factor attained based on our optimized protocol (Our work: ~ 3.92 , Zwettler et al.: ~ 3.4 , Xu et al.: ~ 3), we could achieve a higher effective resolution than the other two works. However, claiming a 3-nm resolution in our Ex-dSTORM might be a misunderstanding. In the manuscript, we initially reported an effectively ~ 3 -nm localization “precision”; hence, the resolution must be greater (suppose that the resolution is calculated by an FWHM equal to 2.35 times of localization precision). We thank reviewer 2 for suggesting performing a Fourier ring correlation analysis to better estimate our system resolution. Consequently, the effective spatial resolution with a combination of dSTORM and expansion microscopy is 5.79 nm (see **Figure R10d**). In the revised manuscript, we have reported the effective resolution instead of a localization precision to avoid confusion (Line 165). In response to the reviewer’s concern that it was not clear for the reader what was already known and what is new, we have recapitulated in the manuscript to show our contributions and new findings, including **1**) proposing the in-situ drift correction for Ex-dSTORM, **2**) achieving a higher expansion factor, **3**) optimizing two-color Ex-dSTORM imaging, **4**) presenting the ultrastructural details of the distal appendages from the base to tip, and **5**) exploring the role of ODF2 as a coordinator for maintaining the nine-fold symmetry of distal appendages. Moreover, although we had cited Zwettler et al.’s work in Line 86 in the original manuscript, we have also cited this paper in Line 122 and added the sentences in Line 121 to clarify what was already known about the re-embedding process in the current manuscript.

2. The authors claim that C2CD3 labeling displays a 9-fold symmetry in the lumen of the centriole wall. This result is partly consistent with the authors’ own data previously published (Yang et al 2018), since in this paper the authors reports that C2CD3 localize compactly in the centriolar lumen as observed by STORM microscopy. Recently, Figure 6 of the Gaudin and colleagues (2022) shows an asymmetric labeling of C2CD3 in the centriole lumen on expanded centriole. How do the authors reconcile this discrepancy? In any case, the Gaudin paper should be cited.

Reply: We would like to thank the reviewer for pointing out the reported observation on the asymmetric labeling of C2CD3 in the recent work (Gaudin et al., 2022). It is advised that we should mention this finding and explain the discrepancy between our data and that of Gaudin and colleagues (2022). In their report, following the protocol of ultra-expansion microscopy (U-ExM), it is evident to find a signal depletion on a particular part in the ring of C2CD3. Nevertheless, in our studies, we obtain images of C2CD3 in the shape of the full ring, which contradicts their results. In order to resolve this discrepancy, we scrutinized all our results of C2CD3 again in Ex-dSTORM imaging and its corresponding ExM imaging. Here, two interesting patterns of C2CD3 could be categorized– full ring and intensity asymmetry (**Figure R4**). Special note is given to the images

of the intensity asymmetry part. In our ExM images, we also observed asymmetric labeling as that in Gaudin’s paper. However, in the corresponding Ex-dSTORM images, we could still observe the nine-fold symmetric pattern of C2CD3 with some comparatively weak clusters. This phenomenon could be further disclosed in the saturated Ex-dSTORM images. Typically, in dSTORM, the optical setup couples with the electron multiplication charge-coupled devices (EMCCD) to detect low-intensity light sources, making dSTORM an excellent imaging tool that allows single-molecule detection. Even in a seemingly incomplete ring, molecules in the depletion region can still be revealed with low localization intensity. Based on this, we deemed that the asymmetry labeling of C2CD3 previously proposed is more likely due to some varying amount of localization within nine C2CD3 puncta than the missing nine-fold symmetric distribution. We have cited the work by Gaudin et al. and added sentences in the revised manuscript (Line 190, **Supplementary Note 3**, and **Supplementary Figure 6**) to clarify the discrepancy.

Figure R4 Intensity Asymmetry of C2CD3. Imaging results of C2CD3 were categorized into the full ring and intensity asymmetry parts by distinct morphologies displayed in the ExM images. Following the group of intensity asymmetry, the images of Ex-dSTORM were further processed to allow the single-molecule signals to become saturated and lead to the revelation of low-intensity signals. Asterisks denote the weak signals that may result in intensity asymmetry. Scale bar, 100 nm.

3. In the title of the paper, the authors mention an “ultrastructural root constitution of distal appendages”. It is not clear to me what the authors are thinking by “root”. I interpret that “root” means the requirement of the root protein to drive distal appendage assembly. Therefore, the

delocalization of the root protein should recruit appendage proteins. In this case the experiment should be performed.

Reply: In response to the reviewer's concern about the title of the paper, we initially used the "root constitution of distal appendages" to designate the proteins or distal appendage-associated proteins that spatially localize close to the base of distal appendages different from those components (CEP83, CEP89, SCLT1, CEP164, and FBF1) occupying at the outer area of the distal appendages. Indeed, it seems inadequate to use "root" in the title if the data was only based on spatial information or if no further functional study, for instance, an experiment on the delocalization of the root protein, was provided. Therefore, We have modified the title to avoid confusion by removing the "root." As suggested by the reviewer, we have performed additional experiments to examine the role of OFD2 in the distal appendage structure (details in the #4 question below).

4. It is intriguing that the authors define ODF2 as the root protein, since the depletion of ODF2 in RPE1 (Tanos et al, 2013; Kuhns et al 2013 and Viol et al 2020) does not affect the presence of distal appendages as observed by electron microscopy. If my interpretation of root is correct, I would suggest that the authors deplete ODDF2 and look at the assembly of distal appendages with their own cell line (RPE1) which might be different from the other teams.

Reply: Although in our study, the distal-layered ODF2 was revealed to correlate highly with distal appendage proteins, we have acknowledged that it was not evident to regard it as the root structure of the distal appendages. Hence, further Ex-dSTORM imaging of the proteins upon ODF2 depletion (clonal RPE-1 cell lines permanently depleted of ODF2 by the CRISPR/Cas 9 method³) was performed and coupled to identify the possible structural alterations in both WT and ODF2^{-/-} cells (**Figure R5a**). In contrast to previous studies where genetic analyses of changes in DAP in either siODF2 or ODF2^{-/-} cells were mainly achieved by traditional fluorescence microscopy⁴⁻⁶, our imaging method provides a molecular-scaled resolution that allows delicate changes to be more discernable. As shown in **Figure R5a**, we demonstrated five single-color Ex-dSTORM images of nine selected proteins in both WT and ODF2^{-/-} cells. Interestingly, in the ODF2^{-/-} cells, several proteins show that puncta are missing at some locations of the iconic nine-fold symmetric arrangement (**Figure R5a**, magenta arrowhead indicates the missing puncta). Out of these proteins that displayed at least a circular pattern, FBF1 was the most prominent structure that was grossly impacted and even deemed unidentifiable in the single-color images. Subsequently, intrigued by this discovery, we performed the two-color Ex-dSTORM imaging of the pair FBF1-SCLT1 in ODF2^{-/-} cells, where SCLT1 was preserved relatively more, to reveal the arrangement of FBF1 and

to inspect its structural defects (**Figure R5b**). We noticed a disordered spatial organization of FBF1

Figure R5 Effect of ODF2 depletion on the structural arrangement of C2CD3 and distal appendage proteins. **a** Representative Ex-dSTORM images of C2CD3 and distal appendage proteins in WT and ODF2^{-/-} cells. The spatial organization of symbolic nine-fold symmetric patterns could be observed in the WT images. Nevertheless, for the ODF2^{-/-} centrioles, some non-ninefold symmetric patterns are revealed where the missing puncta are indicated with arrowheads (magenta). **b** Representative two-color Ex-dSTORM images of FBF1 and SCLT1 in ODF2^{-/-} cells. FBF1 is grossly impacted upon ODF2 depletion, while the typical SCLT1 pattern is mostly retained. Scale bars, 100 nm (**a**, **b**).

proteins, puncta exhibiting at the periphery as well as the center near centriolar triplets. Hence, based on these observations, we conceived that the number of disorganized distal appendage structures is considerably affected by ODF2 depletion. In addition to FBF1, we also noticed that additional signals were displayed at the inner of the CEP164 ring, which was not discovered in WT but in ODF2^{-/-} cells. On the other hand, it is interesting to point out that MNR and CEP90, whose radius is smaller than ODF2, seem not to be affected on their arrangement by ODF2 depletion. Nonetheless, C2CD3, located at the centriolar lumen with an even smaller radius, is found to be frequently lost in one of nine puncta (already checked with saturated image). Last but not least, as the nine-fold symmetry of CEP89 is held in ODF2^{-/-} cells, its charity becomes more evident, which may result from the absence of the proximal layer of CEP89 owing to the subdistal appendage depletion by knocking out ODF2 (Figure R5a, CEP89).

Figure R6 Quantitative analyses of disorganized distal appendage proteins upon ODF2 depletion. **a** Percentage of non-ninefold symmetric distribution of all proteins recorded in ODF2^{-/-} cells (N = 3 independent experiments; at least six centrioles were measured per experiment for C2CD3; at least ten centrioles were measured per experiment for other proteins). **b** Comparison of non-ninefold symmetric pattern of five selected proteins from **a** with the percentage over 1/3, compared across the WT and ODF2^{-/-} cells (N = 3 independent experiments; at least six centrioles were measured per experiment for C2CD3; at least ten centrioles were measured per experiment for other proteins). An interesting remark is given to FBF1, which shows the most considerable fraction of irregular patterns among all other proteins. **c** Comparison of ciliation percentage between WT and ODF2^{-/-} cells. All data are presented as mean \pm SD. Statistical analyses were performed based on two-tailed unpaired t-test individually (** p < 0.01, *** p < 0.001, **** p < 0.0001).

For quantitative analysis, we first calculated the percentage of the non-ninefold symmetric pattern in nine selected proteins upon ODF2 depletion, as shown in **Figure R6a**. Notably, over 80% of FBF1 present non-ninefold symmetric distribution in the ODF2^{-/-} centrioles. Moreover, C2CD3, CEP89, SCLT1, and CEP164 demonstrate over one-third of irregular patterns in ODF2^{-/-} cells. To confirm that this phenomenon is particularly found in the ODF2^{-/-} cells, we compared those proteins exhibiting over one-third of irregular patterns in the ODF2^{-/-} cells to that in WT cells. Surprisingly, all five proteins displayed considerable differences, and FBF1 showed the most significant difference of all ($p < 0.0001$) (**Figure R6b**). Besides inspecting the structural organization, we also compared the ciliation rate (24 h serum withdrawal) between WT and ODF2^{-/-} cells. Conspicuously, the data indicate that the ciliation in ODF2^{-/-} cells is much lower than that in WT cells (**Figure R6c**). These results confirm that ODF2 is not required for the presence of distal appendage. However, ODF2 plays a role in maintaining the exact nine-fold symmetry of the distal appendage structure. Therefore, loss of ODF2 may lead to the irregular localization of distal appendage structure, especially for FBF1, thought of as a gating protein in the distal appendages⁷, and may further affect the ciliogenesis. Hence, we have modified the original claim that ODF2 acted as the root structure of the distal appendage. Instead, our finding suggests that the distal layer of ODF2 plays an auxiliary role of the distal appendage for coordinating and maintaining the distal appendage structure. We have added the results of **Figure R5** and **Figure R6** to the supporting information (**Supplementary Figures 17** and **18**) and included these findings and discussions in the revised manuscript (Line 438-488 and Line 517-519).

5. Two recent papers: Kumar et al 2021 (Journal of cell biology) and LeBorgne et al, 2021 (BioRxiv 2021.07.13.452210) show that the complex containing MNR, CEP90, OFD1 and FOPNL were required for distal appendage assembly. In these two papers, the localization of the different member of this complex has been analyzed using pre- and post- expansion labeling. The results show that these proteins are located between -tubulin or and tubulin used as a reference for the centriolar microtubule wall and the most-proximal distal appendage protein CEP83. Surprisingly the authors do not cite these papers.

Reply: We thank the reviewer for pointing out the important information. In the current revision, we have conducted the Ex-dSTORM imaging of MNR, CEP90, and OFD1 and further performed their localization analyses, including radial, angular and longitudinal distributions. Also, we have cited Kumar et al.'s and LeBorgne et al.'s work in Line 346 to show the previous finding of the gap between CEP83 and Ac-Tub where MNR, CEP90, OFD1, and FOPNL occupy. In **Figure R7a**, the Ex-dSTORM images of MNR, CEP90, and OFD1 manifest an obvious nine-fold symmetry

pattern and their possible chirality. Moreover, we performed two-color Ex-dSTORM imaging of these three proteins with SCLT1 from the top and lateral views to characterize their relative positions (**Figure R7b**). Using SCLT1 as the reference, the relative angular locations of the three proteins are revealed (**Figure R7c**). Furthermore, we have included the radial distribution of MNR, CEP90, and OFD1 to indicate their radial coverage presented with mean radii (**Figure R7d**). Finally, the longitudinal positions of MNR, CEP90, and OFD1 relative to the SCLT1 are demonstrated in **Figure R7e**. The result in **Figure R7** has been added to **Figure 6** and **Supplementary Figure 13** in the revised manuscript to elucidate the molecular mapping of MNR, CEP90, and OFD1 with a resolution of a few nanometers.

Figure R7 Ex-dSTORM images and the corresponding ultrastructural analyses of MNR, CEP90, and OFD1. **a** Two representative Ex-dSTORM images of distal appendage proteins MNR, CEP90, and OFD1. **b** Two-color Ex-dSTORM images of MNR, CEP90, and OFD1 with SCLT1 from the top and lateral views. **c** Schematic arrangements of MNR, CEP90, and OFD1, whose quantitative angular displacements against SCLT1 are indicated as mean values and standard deviations. **d** Radial coverage of MNR, CEP90, and OFD1 together with FBF1, SCLT1, CEP83, CEP89, Ac-Tub and C2CD3 (mean \pm 3 SD for each protein). The quantitative mean radii are marked in the corresponding positions. **e** Histogram analysis of longitudinal positions of MNR, CEP90, and OFD1 with other distal appendage proteins and C2CD3 relative to SCLT1. Scale bars, 100 nm (**a**, **b**)

6. Line 163, the authors claim that C2CD3 is a core distal appendage protein. C2CD3 is required for distal appendage assembly, however its localization in the centriolar lumen prevents this hypothesis as already suggested by the corresponding author of this paper (Yang et al, 2018).

Reply: As mentioned by the reviewer, C2CD3 plays an indispensable role in the distal appendage assembly, which raises an interesting question about whether its localization is correlated with distal appendage proteins. Hence, we conducted Ex-dSTORM imaging of C2CD3 to better characterize its localization. Our results showed a precise nine-fold symmetric distribution of C2CD3 protein, surprisingly localizing to the inner centriole wall (mean radius = 143.9 nm) rather than the central lumen to form a compact ring structure (mean radius = 76 nm) observed in the previous work (Yang et al, 2018) under a dSTORM resolution. In addition to its proximity to the root configuration of distal appendages, we found that C2CD3 shows a high angular correlation with other distal appendage proteins, such as SCLT1. Although C2CD3 is involved in distal appendage assembly, it might not be obvious to claim that C2CD3 belongs to the core appendage proteins because it is at the inner wall of the centriole. Thus, we have modified the statement of C2CD3 as a core distal appendage protein in the main text (Line 175) to address this concern.

7. Line 330, The gap between CEP83 and Ac-tub was already observed using post-labeling expansion microscopy in Gaudin et al, 2022 and Le Borgne et al, BioRxiv, 2021.

Reply: We thank the reviewer for the comment. We have cited these two important recent works to show the observation of the gap between CEP83 and Ac-tub using post-labeling expansion microscopy (Line 346). In this manuscript, we further performed Ex-dSTORM imaging to highlight this specific coverage with additional ten-fold resolution improvement by dSTORM. As mentioned above, we have included the Ex-dSTORM results of MNR, CEP90, and OFD1 proteins in the current manuscript to reveal their protein organization at a molecular scale. We have also modified the paragraph in the main text (Line 342-359) to clarify this.

8. Line 363, "our finding suggests that ODF2 fills the undefined coverage, ...". This statement excludes other proteins already known (such as MNR, FOPNL, CEP90 and OFD1) or to be discovered that localize there.

Reply: In response to the reviewer's comment regarding the statement of ODF2, we have already added the results of MNR, CEP90, and OFD1 to the revised manuscript, which discusses more general protein candidates potentially localizing to the coverage between CEP83 and centriolar triplets. We have modified the sentences in Line 390-393 (revised version) about ODF2, which was also found within this specific region near the base of distal appendages. Further discussion about ODF2's role in the organization of distal appendage proteins has been illustrated in **Figure R5** and **Figure R6**.

9. Line 388, the authors claim that "the morphology of DAP would alter upon sDAP depletion". In their previous paper (Chong et al, 2020), the same conclusion was already reached. I would have expected a comment such as "as previously shown by ..."

Reply: We thank the reviewer for pointing this out. In the previous study (Chong et al., 2020), we found that the gap between ODF2-C/N and SCLT1 in CEP128^{-/-} cells became larger when compared to that in WT cells. In addition, the proximal layer of CEP89 was absent upon CEP128 KO. These changes in distal appendages were observed from the longitudinal positions of these proteins. Nevertheless, in the current work, the conclusion (Line 388 in the original manuscript) was drawn based on the radial analysis observed from the top view, indicating a smaller radius ratio of ODF2-C/N to SCLT in CEP128^{-/-} cells because we regarded ODF2 as a part of distal appendage structure in the original manuscript. Thus, we claimed that "the morphology of the DAP would alter upon sDAP depletion." However, in the revised manuscript, we have further investigated the influence of ODF2 on distal appendages (**Figure R6** and **Figure R6**) and proposed a role of ODF2 as a coordinator, not the root constitution, of the distal appendage structure. Hence, we have removed the previous statement in Line 388 to avoid confusion.

Other comments

10. Line 445-447: The authors talk about a 18° offset between the 2 layers of OFD2. Can the authors explain to which data they refer?

Reply: We reported this angle offset between the two layers of ODF2 in Line 399 (or Line 369 in the original version). The quantitative analysis of this angle is elaborated in **Supplementary Figure 15** (or **Supplementary Figure 12** in the original supporting information).

11. Some scale bars are missing.

Reply: In the current revision, we have added scale bars in **Fig. 4b-e** and **Fig. 8a**.

12. The representation of the measured angle could have been done in an easier way.

Reply: The angular analysis to determine the relative radial direction among proteins was detailed in **Supplementary Figures 8 and 9** (revised version). At first glance, the representation of the measured angle seems not straightforward because relative angles ranging over multiple intervals between two puncta in a nine-fold symmetric pattern were measured. However, this analysis does possess advantages. For instance, the relative angle was fairly estimated due to the more data points used to yield a reliable mean value. A meaningful relative angle would be found as it was supposed to repeat at the same angular positions of each interval (every 40 degrees) (see **Supplementary Figure 9**). Moreover, we can validate results by examining how the angle histogram in each interval is correlated through this robust analysis. This method also indicates the direction of angular displacements between two proteins to show their relative organization. Thus, we think the presented method is necessary for angular measurement.

13. Material and methods could have been better explained as well as figure legends

Reply: We have included more detailed information in the methods section as well as the figure legends for a better explanation of the in-situ drift correction process, expansion protocol, re-embedding process, Ex-dSTORM imaging, cell lines, reagents, etc.

Reviewer #2 (Remarks to the Author):

The manuscript used Ex-dSTORM to study the molecular organization of centrioles with a particular focus on the root structure of centriolar DAP in cells. After ~4-fold expansion in charged polyacrylamide gel the sample was re-embedded to avoid shrinking of the gel in photoswitching buffer. Using an optimized re-embedding protocol the authors achieved a final expansion factor of 3.9, i.e. slightly larger than shown in a previous Ex-dSTORM paper. While previous works have shown 3D Ex-dSTORM with a single color, the present work used two-color 2D to map the position of different immunolabeled proteins. Performing frontal and side view 2D dSTORM images the authors could finally reveal detailed information about the molecular arrangement of DAP proteins. I recommend acceptance of the manuscript in Nat Commun but request the authors to tone down a few statements that are as presented untenable.

Reply: We appreciate the positive feedback from the reviewer. For the biological part, we claimed that ODF2 proteins might work as the root structure of the distal appendage in the original manuscript. Nevertheless, it seems not evident if no functional study was provided or if the conclusion was drawn only based on ODF2's localization near the root of the distal appendage. To gain an insight into the role of ODF2 in distal appendages, we further explored the molecular localization of several distal appendage proteins in the ODF2-depleted centrioles using Ex-dSTORM. The result shown in the revised manuscript suggests that ODF2 is a coordinator for maintaining the nine-fold symmetric distribution of some distal appendage proteins. Thus, we have modified the statements regarding ODF2's role and provided new evidence to show its effect on distal appendage structure (Line 438-488 and **Supplementary Figures 17 and 18**).

For the resolution described in the original manuscript, we claimed a 2~3 nm localization precision, which was obtained by dividing the measured localization precision (~11 nm) by the expansion factor of 3.92. As advised by the reviewer, this resolution estimation was inaccurate since the localization precision was still ~11 nm due to the same photon budget detected. Hence, we have modified the statement in Line 163-167, shown in the revised manuscript to correct this point. Besides, we have performed FRC analysis to evaluate a realistic resolution of our Ex-dSTORM images. As a result, an effective resolution achieved by combining expansion microscopy and dSTORM is provided (Line 168 and **Supplementary Figure 5**).

1. *A previous Ex-dSTORM study of centrioles resolved at least partially the microtubule triplets as hollow structures (Ref. 41). In the present manuscript the tubulin signal (e.g. Fig. 2) indicates a substantially lower spatial resolution, why? Can the authors comment of that?*

Reply: In response to the reviewer’s concern about our spatial resolution, the Ex-dSTORM images of tubulin signals in the original manuscript were to present the microtubule triplets (**Fig. 2a**), which were not possibly resolved by dSTORM alone. A rendering pixel size of 11.625 nm was usually used for most of our Ex-dSTORM images. Nevertheless, a finer pixel size may be needed to show more details if we investigate smaller features, for instance, the hollow structure of microtubule triplets. Therefore, we re-rendered the image by increasing the sampling factor by 1.5 folds finer. The result is shown in **Figure R8**. Similarly, we found some hollow structures of the microtubule triplets stained for Ac-tub were resolved (**Figure R8a and b**). The dashed boxes in **Figure R8a** mark the possible hollow structures of microtubule triplets, demonstrating the ring-like pattern.

However, imaging quality might be compromised because the signals collected within several hundreds of nanometers in the z-axis direction may hinder the precise imaging of the hollow structure of microtubule triplets that especially twist along the longitudinal direction of the centriole. Although we achieve a higher expansion factor, the extended distance (height) of the mammalian centrioles from the coverslip would compromise the signal-to-noise level compared to the isolated *Chlamydomonas* centrioles placed on the coverslips (Zwettler et al.’s work, Ref. 41).

Figure R8 Resolving the hollow structure of microtubule triplets. a Ex-dSTORM imaging with a finer rendering pixel size for revealing more details of microtubule triplets (stained with Ac-Tub in RPE-1 cells). The green dashed boxes mark the resolved hollow structures of the microtubule triplets. **b** Enlarged images of the marked regions in **a**. The ring-like pattern of centriolar triplets illustrates the hollow structure of microtubules. Scale bar, 100 nm (**a**), 10 nm (**b**).

2. In the introduction the authors state that they will introduce a method that enables Ex-dSTORM imaging throughout the cell but then they use TIRF illumination with an oil-immersion objective. This needs more explanation.

Reply: We apologize for raising some confusion. In practice, the oil immersion objective (100X 1.49, CFI Apo TIRF, Nikon) was used for epi-illumination, where the entire depth of the sample was illuminated. In the original manuscript, the information on wide-field illumination was mentioned in the methods section for Ex-dSTORM imaging in Line 561. Although **Fig. 1b** illustrated the epi-illumination scheme, no clear label was shown. Thus, we have added the label “epi-illumination” in **Fig. 1b**, as shown in **Figure R9** and described the illumination configuration in the methods section (Line 663-664, Ex-dSTORM imaging, Methods).

Figure R9 Epi-illumination for in-situ drift correction enabling Ex-dSTORM imaging of target proteins. The microsphere beads deposited on the coverslip no longer work as fiducial markers in the expanded sample due to a long distance from the imaging plane. Furthermore, epi-illumination is adopted in the optical setup to allow the entire depth of sample illumination.

3. Even though the authors use TIRF they achieve only a localization precision of ~ 12 nm, i.e. lower than in other dSTORM studies. This corresponds to a spatial resolution of ~ 30 nm. Hence, with 4-fold expansion the method can achieve ~ 8 nm spatial resolution (best case scenario). But, a spatial resolution of ~ 10 nm is still far away from a resolution well close to an EM level as claimed in the intro. I request the authors to perform FRC on their images or another resolution estimation to come up with realistic resolution claims.

Furthermore, the localization precision remains at 12 nm even in the expanded sample. It is wrong to divide the loc precision with the expansion factor and then state 3 nm loc precision.

Reply: We would like to thank the reviewer for the detailed comments and suggestions. In order to report the realistic resolution of our Ex-dSTORM images, the resolution estimation is based on

the target protein at the centriole of a cell, usually not close to coverslips. Thus, we expected a lower contrast in single-molecule imaging, which slightly upset the spatial resolution. In order to obtain a realistic resolution claim, we have performed analytical measures on the representative protein, Ac-tub shown in **Figure R10**. Following the imaging result (**Figure R10a**), we first analyzed the acquired image in terms of the localization precision, which was 11.50 ± 4.07 nm (**Figure R10b**). Then, as instructed, we further performed FRC analysis. Here, since the FRC cutoff returns 0.044 nm^{-1} , the resolution, the reciprocal of the FRC cutoff, equals 22.71 nm (**Figure R10c**). Finally, we compiled the information all together in **Figure R10d**. Further division of the average expansion factor gives the effective spatial resolution equal to 5.79 nm. In the revised version, we have used “effective” resolution to avoid the misunderstanding of improving resolution in our dSTORM imaging. In the supporting information, we provided both analyses in the revision for reference (**Supplementary Figure 5**). Furthermore, we have removed the related resolution claim of “2-3 nm localization precision”. More importantly, the resolution obtained here is still a few distance away from EM level. Hence, we have removed the resolution claim related to the EM.

* 3.92 is the averaged expansion factor in this work.

Figure R10 Validation of resolution. **a** Representative Ex-dSTORM image of Ac-Tub in RPE-1 cell for the resolution analysis. **b** Histograms analysis of localization precision from single molecules per switching event. **c** FRC analysis on the image shown in **a** with a threshold designated to $1/7$, as indicated in the blue line. **d** Table of resolution data from **b** and **c** along with an additional row of effective resolution value incorporated with the expansion factor gained in this study. Scale bar (without scaled), 500 nm (**a**)

4. Finally, I do not understand why the authors tested ATTO647N for dSTORM imaging since it is known that ATTO47N does not show photoswitching in thiol-buffer. In addition, it has been shown that ATTO647N shows dramatic photobleaching (Ref. 46).

Reply: Regarding the reviewer's comment, we refer to Dempsey et al.'s paper. The work reports that ATTO647N shows low to moderate photoswitching property in thiol-buffer (MEA) together with GLOX (oxygen scavenging system) even though thiol or GLOX alone does not show photoswitching (see red box in **Figure R11**⁸). Also, ATTO647N presents moderate sensitivity to violet-light photoactivation, which helps regulate the density of single-molecule signals (green box, **Figure R11**). Despite having a better photoswitching property from ATTO647N in the thiol buffer using β -mercaptoethanol (β ME) with GLOX, we still used MEA and GLOX for a fair comparison with AF647 and Dy654. Moreover, our result indicates that ATTO647N exhibits an acceptable labeling quality in the expanded cells. As mentioned by the reviewer, Helmerich and colleagues⁹ (Ref. 46) have reported that the ATTO647N showed a more dramatic photobleaching effect than the Alexa 647. This discrepancy between Helmerich et al.'s and our findings may be caused by the capturing intensity. We have included Helmerich's results and added the imaging condition in each

Table 2 | Sensitivity to violet-light photoactivation

Dye	Sensitivity ^a	
Blue-absorbing	Atto 488	+
	Alexa Fluor 488	+
	Atto 520	+
	Fluorescein	-
Yellow-absorbing	FITC	-
	Cy2	-
	Cy3B	+
	Alexa Fluor 568	+
	TAMRA	-
	Cy3	-
Red-absorbing	Cy3.5	+
	Atto 565	+
	Alexa Fluor 647	++
	Cy5	++
	Atto 647	+
	Atto 647N	+
	Dyomics 654	++
	Atto 655	+
NIR-absorbing	Atto 680	+
	Cy5.5	++
	DyLight 750	++
	Cy7	++
	Alexa Fluor 750	++
	Atto 740	+
Alexa Fluor 790	++	
IRDye 800CW	++	

^aFraction of single fluorescent molecules that activate after a 0.25 s pulse of 405 nm excitation light (10–30 W cm⁻²) relative to the total initial population was measured for each dye. Dyes were assigned a "-", "+", or "++" if <3%, 3–25% or >25% were reactivated, respectively.

Dempsey, Graham T., et al. "Evaluation of fluorophores for optimal performance in localization-based super-resolution imaging." *Nature methods* 8.12 (2011): 1027-1036.

Table 3 | Effect of buffer composition on photoswitching properties

Dye		Buffer condition			
		No GLOX or thiol	Thiol only	GLOX only	GLOX and thiol
Blue-absorbing	Atto 488	+	+	++	++
	Alexa Fluor 488	+	+	++	++
	Atto 520	+	+	++	++
	Fluorescein	-	+	+	+
	FITC	-	+	+	+
	Cy2	-	-	-	-
Yellow-absorbing	Cy3B	+	+	++	++
	Alexa Fluor 568	-	-	+	+
	TAMRA	-	-	-	+
	Cy3	-	-	+	+
	Cy3.5	-	-	+	+
	Atto 565	-	-	+	+
Red-absorbing	Alexa Fluor 647	-	+	-	++
	Cy5	-	+	-	++
	Atto 647	+	+	+	+
	Atto 647N	+	-	-	+
	Dyomics 654	-	+	-	++
	Atto 655	+	+	+	++
NIR-absorbing	Atto 680	+	+	++	++
	Cy5.5	-	-	-	+
	DyLight 750	-	+	-	++
	Cy7	-	+	-	++
	Alexa Fluor 750	-	+	-	+
	Atto 740	-	+	+	+
Alexa Fluor 790	-	+	-	+	
IRDye 800CW	-	-	-	-	

Cells were immunostained for microtubules using each of the dyes. The thiol used was MEA. Dyes were assigned a "-", "+", or "++" if rapid bleaching (no image), low to moderate photoswitching (low to moderate quality image) or robust photoswitching (high quality image) was observed, respectively. Whereas lower image quality (+) was observed for Atto 647N and Alexa Fluor 750 in the 'GLOX and thiol' condition, reasonable image quality (++) was observed for these two dyes when β ME was used instead of MEA.

Figure R11⁸ Previous report showing photoswitching property of ATTO647N. Dempsey et al. demonstrated the low to moderate photoswitching property of ATTO647N with MEA and GLOX together, as indicated in the red box. Moreover, the green box marks its moderate sensitivity to violet-light photoactivation.

image panel shown in **Figure R12**. Notably, the imaging condition described for panel **d** is compared with that used in our crosstalk analysis. They showed that irradiation of 641 nm for 1 min with 2 kW cm^{-2} intensity was done before capturing the image in the shorter wavelength channel with an illumination intensity of 180 W cm^{-2} much lower than that for dSTORM imaging acquisition (typically $2\text{-}5 \text{ kW cm}^{-2}$ in our experiments). This difference in laser intensity may lead to inconsistent results between two works. One possible reason is that although ATTO647N possesses high photobleaching conversion, its converted state might present a much lower survival fraction of single-molecule blinking compared to Alexa 647 at a high intensity of 561 nm illumination. This may explain why using ATTO647N still poses relatively lower crosstalk in the two-color SMLM imaging. Indeed, owing to this interesting discrepancy, we would like to introduce ATTO647N for comparison of the crosstalk experiment; this may raise additional important information about the underlying mechanism to be explored, which relates photobleaching to the crosstalk phenomenon at different radiation intensities, imaging buffers, etc.

Helmerich, Dominic A., et al. "Photobleaching of organic dyes can cause artifacts in super-resolution microscopy." *Nature Methods* 18.3 (2021): 253-257.

Figure R12⁹ Comparison of the imaging conditions in Helmerich et al.'s work on the photobleaching effect. We added the corresponding imaging conditions for **a** through **d**. Special note is given to **d**; the irradiation at 647 nm with 2 kW cm⁻² intensity before capturing is compared to our sequential two-color dSTORM imaging. The illumination intensity for capturing an image in **d** is much lower than that used in our dSTORM imaging (typically 2-5 kW cm⁻²).

Additional information:

In our original manuscript, we made a mistake when defining the observed orientation of the distal appendage; this was due to flipped images caused during image acquisition. The claim that the centrioles were viewed from the distal end was incorrect. They were actually observed from the proximal end. Therefore, we have corrected the results related to this issue.

References:

1. Zwettler FU, et al. Molecular resolution imaging by post-labeling expansion single-molecule localization microscopy (Ex-SMLM). *Nature communications* 11, 1-11 (2020).
2. Chen F, et al. Nanoscale imaging of RNA with expansion microscopy. *Nature methods* 13, 679-684 (2016).
3. Mazo G, Soplop N, Wang W-J, Uryu K, Tsou M-FB. Spatial control of primary ciliogenesis by subdistal appendages alters sensation-associated properties of cilia. *Developmental cell* 39, 424-437 (2016).
4. Tanos BE, et al. Centriole distal appendages promote membrane docking, leading to cilia initiation. *Genes & development* 27, 163-168 (2013).
5. Viol L, et al. Nek2 kinase displaces distal appendages from the mother centriole prior to mitosis. *Journal of Cell Biology* 219, (2020).
6. Kuhns S, et al. The microtubule affinity regulating kinase MARK4 promotes axoneme extension during early ciliogenesis. *Journal of Cell Biology* 200, 505-522 (2013).
7. Yang TT, et al. Super-resolution architecture of mammalian centriole distal appendages reveals distinct blade and matrix functional components. *Nature communications* 9, 1-11 (2018).
8. Dempsey GT, Vaughan JC, Chen KH, Bates M, Zhuang X. Evaluation of fluorophores for optimal performance in localization-based super-resolution imaging. *Nature methods* 8, 1027-1036 (2011).
9. Helmerich DA, Beliu G, Matikonda SS, Schnermann MJ, Sauer M. Photobleaching of organic dyes can cause artifacts in super-resolution microscopy. *Nature Methods* 18, 253-257 (2021).

REVIEWERS' COMMENTS

Reviewer #1 (Remarks to the Author):

The authors have largely improved their manuscript based on the comments of the first round of reviews and provide new interesting data on the fate of the all the known proteins required for distal appendages assembly. Concerning my specific comments, the authors have addressed them sufficiently well.

However, I still have a problem concerning the resolution. It is still not clear why the authors divide by 4, the expansion factor, the resolution obtained by the Fourier ring correlation. In my opinion, the image resolution is the one that it is obtained by FRC analysis. Can the authors be more precise concerning this point. In addition can the authors precise which ImageJ plugin they are using. In Supplementary Fig 17, it seems that the two-colors Ex-dSTORM imaging of the pair FBF1-SCLt1 is more disorganized that the images in the simple color. Do the authors have comments about this point? Nevertheless, I am convinced that ODF2^{-/-} cells show missing dot for several proteins and the feather structure of CEP164 appears highly disorganized.

Reviewer #2 (Remarks to the Author):

The authors addressed all my concerns. I am happy to recommend the revised manuscript for publication in Nature Communications.

Reviewer #1 (Remarks to the Author):

The authors have largely improved their manuscript based on the comments of the first round of reviews and provide new interesting data on the fate of the all the known proteins required for distal appendages assembly. Concerning my specific comments, the authors have addressed them sufficiently well.

However, I still have a problem concerning the resolution. It is still not clear why the authors divide by 4, the expansion factor, the resolution obtained by the Fourier ring correlation. In my opinion, the image resolution is the one that it is obtained by FRC analysis. Can the authors be more precise concerning this point. In addition can the authors precise which ImageJ plugin they are using.

Reply: We appreciate the positive feedback from the reviewer. In response to the reviewer's concern about the reported resolution that we divide by the expansion factor, the actual resolution of our dSTORM is indeed the one from the FRC analysis. Therefore, any features with a peak-to-peak distance smaller than our dSTORM resolution, i.e., ~22.7 nm, can not be resolved. In this manuscript, we performed dSTORM imaging for the expanded cells. With the combination of two techniques, two fluorophores initially separated by a distance of more than 5.8 nm (22.7 divided by the expansion factor) can still be differentiated because the distance between these two fluorophores is more than 22.7 nm after the expansion procedure. That is why our Ex-dSTORM provides a higher resolving power than dSTORM alone. Thus, our Ex-dSTORM images show more detailed molecular localizations compared to dSTORM. To clarify this and avoid confusion, we have changed the “effective resolution” shown in the previous manuscript to “the shortest distance between two labeled molecules that can be distinguished with Ex-dSTORM.” For the FRC analysis, we used the GDSC SMLM, an ImageJ-based plugin (Etheridge et al., 2022), to evaluate our system resolution. We have added this information to the method section.

Reference:

Etheridge, Thomas J., Antony M. Carr, and Alex D. Herbert. “GDSC SMLM: Single-molecule localization microscopy software for ImageJ.” *Wellcome Open Research* 7 (2022): 241.

In Supplementary Fig 17, it seems that the two-colors Ex-dSTORM imaging of the pair FBF1-SCLt1 is more disorganized than the images in the simple color. Do the authors have comments about this point? Nevertheless, I am convinced that ODF2^{-/-} cells show missing dot for several proteins and the feather structure of CEP164 appears highly disorganized.

Reply: We would like to thank the reviewer for pointing this out. For the single-color result, we specifically searched for the ring-shaped pattern for the DAP and DAP-associated proteins to ensure that all signals we imaged were truly from the target proteins since only a single protein was labeled in each sample. Therefore, the single-color images reported in Supplementary Fig. 17 are prone to organized patterns due to the restricted search criterion used for centriole recognition. The same criterion was applied for both WT and ODF2^{-/-} cells; thus, the comparison is still valid. Nevertheless, for the two-color images, we first identified the DAP signals based on the SCLT1 pattern because SCLT1 was relatively organized. Then, we acquired FBF1 images accordingly. In this scenario, there was no preset search condition for FBF1. So, a more disorganized pattern was obtained in FBF1 images.